# Provably Efficient Reinforcement Learning with Multinomial Logit Function Approximation

**Long-Fei Li**[1,2], **Yu-Jie Zhang**[3], **Peng Zhao**[1,2], **Zhi-Hua Zhou**[1,2]

[1] National Key Laboratory for Novel Software Technology, Nanjing University, China
[2] School of Artificial Intelligence, Nanjing University, China
[3] The University of Tokyo, Chiba, Japan

## Abstract

We study a new class of MDPs that employs multinomial logit (MNL) function approximation to ensure valid probability distributions over the state space. Despite its significant benefits, incorporating the non-linear function raises substantial challenges in both *statistical* and *computational* efficiency. The best-known result of Hwang and Oh [2023] has achieved an $\widetilde{\mathcal{O}}(\kappa^{-1}dH^2\sqrt{K})$ regret upper bound, where $\kappa$ is a problem-dependent quantity, $d$ is the feature dimension, $H$ is the episode length, and $K$ is the number of episodes. However, we observe that $\kappa^{-1}$ exhibits polynomial dependence on the number of reachable states, which can be as large as the state space size in the worst case and thus undermines the motivation for function approximation. Additionally, their method requires storing all historical data and the time complexity scales linearly with the episode count, which is computationally expensive. In this work, we propose a statistically efficient algorithm that achieves a regret of $\widetilde{\mathcal{O}}(dH^2\sqrt{K} + \kappa^{-1}d^2H^2)$, eliminating the dependence on $\kappa^{-1}$ in the dominant term for the first time. We then address the computational challenges by introducing an enhanced algorithm that achieves the same regret guarantee but with only constant cost. Finally, we establish the first lower bound for this problem, justifying the optimality of our results in $d$ and $K$.

## 1 Introduction

Reinforcement Learning (RL) with function approximation has achieved remarkable success in various applications involving large state and action spaces, such as games [Silver et al., 2016], algorithm discovery [Fawzi et al., 2022] and large language models [Ouyang et al., 2022]. Therefore, establishing the theoretical foundation for RL with function approximation is of great importance. Recently, there have been many efforts devoted to understanding the linear function approximation, yielding numerous valuable results [Yang and Wang, 2019, Jin et al., 2020, Ayoub et al., 2020].

While these studies make important steps toward understanding RL with function approximation, there are still challenges to be solved. In linear function approximation, transitions are assumed to be linear in feature mappings, such as $\mathbb{P}(s'|s,a) = \phi(s'|s,a)^\top\theta^*$ for linear mixture MDPs and $\mathbb{P}(s'|s,a) = \phi(s,a)^\top\mu^*(s')$ for linear MDPs. Here $\mathbb{P}(s'|s,a)$ is the probability from state $s$ to $s'$ taking action $a$, $\phi(s'|s,a)$ and $\phi(s,a)$ are feature mappings, $\theta^*$ and $\mu^*(s')$ are unknown parameters. However, the transition function is a *probability distribution* over states, meaning its values must lie within $[0,1]$ and sum to 1. Thus, the linearity assumption is restrictive and hard to satisfy in practice. An algorithm designed for linear MDPs could break down entirely if the underlying MDP is not linear [Jin et al., 2020]. While some works explore generalized linear [Wang et al., 2021] and general function approximation [Russo and Roy, 2013, Foster et al., 2021, Chen et al., 2023], they focus on function approximation for value functions rather than transitions, hence do not tackle this challenge.

---

Correspondence: Peng Zhao <zhaop@lamda.nju.edu.cn>

38th Conference on Neural Information Processing Systems (NeurIPS 2024).

Table 1: Comparison between previous works and ours in terms of the regret and computational cost (including storage and time complexity) . Here $\kappa$ and $\kappa^*$ are problem-dependent quantities defined in Assumption 1, $d$ is the feature dimension, $H$ is the episode length and $K$ is the number of episodes. The computational cost per episode only highlight the dependence on the episode count $k$.

| Reference | Regret | Storage | Time | MDP model |
|---|---|---|---|---|
| Hwang and Oh [2023] | $\widetilde{\mathcal{O}}(\kappa^{-1}dH^2\sqrt{K})$ | $\mathcal{O}(k)$ | $\mathcal{O}(k)$ | homogeneous |
| UCRL-MNL-LL (Theorem 1) | $\widetilde{\mathcal{O}}(dH^2\sqrt{K} + \kappa^{-1}d^2H^2)$ | $\mathcal{O}(k)$ | $\mathcal{O}(k)$ | inhomogeneous |
| UCRL-MNL-OL (Theorem 2) | $\widetilde{\mathcal{O}}(dH^2\sqrt{K} + \kappa^{-1}d^2H^2)$ | $\mathcal{O}(1)$ | $\mathcal{O}(1)$ | inhomogeneous |
| Lower Bound  (Corollary 1) | $\Omega(dH\sqrt{K\kappa^*})$ | – | – | infinite action space |

Towards addressing the limitation of linear function approximation, a new class of MDPs that utilizes multinomial logit function approximation has been proposed by Hwang and Oh [2023] recently. Such formula also aligns better with models like neural networks [LeCun et al., 2015], which inherently respect the probabilistic constraints through a softmax layer and allow for greater expressive power. However, though it offers promising benefits, the introduction of non-linear functions introduces significant challenges in both *statistical* and *computational* efficiency. Specifically, the best-known approach of Hwang and Oh [2023] has achieved an $\widetilde{\mathcal{O}}(\kappa^{-1}dH^2\sqrt{K})$ regret, where $\kappa$ is a problem-dependent quantity that measures the effective non-linearity over the entire parameter space, $d$ is the feature dimension, $H$ is the episode length, and $K$ is the number of episodes. Unfortunately, as we show in Claim 1, it holds that $\kappa^{-1} > U^2$, where $U$ denotes the maximum number of reachable states, which equals to the size of the state space $S$ in the worst case. This undermines the core motivation for function approximation, which aims to mitigate dependence on large state and action spaces. Furthermore, the method requires storing all historical data, and its time complexity *per episode* grows *linearly* with the episode count (i.e., $\mathcal{O}(k)$ at episode $k$). Thus, a natural question arises:

*Is it possible to design both statistically and computationally efficient algorithms for RL with MNL function approximation?*

In this work, we answer this question affirmatively for the class of MNL mixture MDPs where the transition is parameterized by a multinomial logit function. Our contributions are listed as follows:

- For statistical efficiency, we propose the UCRL-MNL-LL algorithm, which attains a regret bound of $\widetilde{\mathcal{O}}(dH^2\sqrt{K} + \kappa^{-1}d^2H^2)$. Our result significantly improves upon the $\widetilde{\mathcal{O}}(\kappa^{-1}dH^2\sqrt{K})$ rate of Hwang and Oh [2023], making the first time to achieve a $\kappa$-independent dominant term (note that the lower-order term still scales with $\kappa^{-1}$, but does not depend on $K$, making it acceptable). To achieve this, we propose a tighter confidence set based on a new Bernstein-type concentration [Périvier and Goyal, 2022] instead of the standard Hoeffding-type concentration, and exploit the self-concordant-like property [Bach, 2010] of the log-loss function to better use local information.

- For computational efficiency, we propose the UCRL-MNL-OL algorithm, which enjoys the same regret bound as UCRL-MNL-LL, but with only *constant* storage and time complexity per episode. This is enabled by recognizing that the negative log-likelihood function is exponentially concave, which motivates the use of online mirror descent with a specifically tailored local norm [Zhang and Sugiyama, 2023] to replace the standard maximum likelihood estimation. Furthermore, we construct the optimistic value function by incorporating a closed-form bonus term through a second-order Taylor expansion, thus avoiding the need to solve a non-convex optimization problem.

- We establish the *first* lower bound for MNL mixture MDPs by introducing a reduction to the logistic bandit problem. We prove a problem-dependent lower bound of $\Omega(dH\sqrt{K\kappa^*})$ for infinite action setting, where $\kappa^*$ is an another problem-dependent quantity that measures the effective non-linearity over around the ground truth parameter. Though this does not constitute a strict lower bound for the finite action case studied in this work, it suggests that our result may be optimal in $d$ and $K$. [1]

Table 1 provides a comparison between our work and previous studies, focusing on regret and computational costs, including both storage and time complexity.

---

[1]After the submission of our work to arXiv [Li et al., 2024a], a follow up work by Park et al. [2024] proved a lower bound of $\Omega(dH^{3/2}\sqrt{K})$ for the finite action setting. This confirms that our result is optimal in $d$ and $K$.

**Organization.** We introduce the related work in Section 2 and present the setup in Section 3. Then, we design a statistically efficient algorithm in Section 4. Next, we present an algorithm that achieves both statistical and computational efficiency in Section 5. Finally, we establish the lower bound in Section 6. Section 7 concludes the paper. Due to space limits, we defer all proofs to the appendixes.

**Notations.** We use $[x]_{[a,b]}$ to denote $\min(\max(x, a), b)$. For a vector $\mathbf{x} \in \mathbb{R}^d$ and positive semi-definite matrix $A \in \mathbb{R}^{d \times d}$, denote $\|\mathbf{x}\|_A = \sqrt{\mathbf{x}^\top A \mathbf{x}}$. For a strictly convex and continuously differentiable function $\psi : \mathcal{W} \mapsto \mathbb{R}$, the Bregman divergence is defined as $\mathcal{D}_\psi(\mathbf{w}_1, \mathbf{w}_2) = \psi(\mathbf{w}_1) - \psi(\mathbf{w}_2) - \langle \nabla\psi(\mathbf{w}_2), \mathbf{w}_1 - \mathbf{w}_2 \rangle$. We use the notation $\mathcal{O}(\cdot)$ to indicate different types of dependencies depending on the context. For regret analysis, $\mathcal{O}(\cdot)$ omits only constant factors. For computational costs, we use $\mathcal{O}(\cdot)$ to solely highlight the dependence on the number of episode as this is the primary factor influencing the complexity. Additionally, we employ $\widetilde{\mathcal{O}}(\cdot)$ to hide all polylogarithmic factors.

## 2 Related Work

In this section, we review related works from both setup and technical perspectives.

**RL with Generalized Linear Function Approximation.** There are recent efforts devoted to investigating function approximation beyond the linear models. Wang et al. [2021] investigated RL with generalized linear function approximation. Notably, unlike our approach that models transitions using a generalized linear model, they apply this approximation directly to the value function. Another line of works [Chowdhury et al., 2021, Li et al., 2022, Ouhamma et al., 2023] has studied RL with exponential function approximation and also aimed to ensure that transitions constitute valid probability distributions. The MDP model can be viewed as an extension of bilinear MDPs in their work while our setting extends linear mixture MDPs. These studies are complementary to ours and not directly comparable. Moreover, these works also enter the computational and statistical challenges arising from non-linear function approximation that remain to be addressed. The most relevant work to ours is the recent work by Hwang and Oh [2023], which firstly explored a similar setting to ours, where the transition is characterized using a multinomial logit model. We significantly improve upon their results by providing statistically and computationally more efficient algorithms.

**RL with General Function Approximation.** There have also been some works that studies RL with general function approximation. Russo and Roy [2013] and Osband and Roy [2014] initiated the study on the minimal structural assumptions that render sample-efficient learning by proposing a structural condition called Eluder dimension. Recently, several works have investigated different conditions for sample-efficient interactive learning, such as Bellman Eluder (BE) dimension [Jin et al., 2021], Bilinear classes [Du et al., 2021], Decision-Estimation Coefficient (DEC) [Foster et al., 2021], and Admissible Bellman Characterization (ABC) [Chen et al., 2023]. A notable difference is that they impose assumptions on the value functions while we study function approximation on the transitions to ensure valid probability distributions. Moreover, the goal of these works is to study the conditions for sample-efficient reinforcement learning, but not focus on the computational efficiency.

**Multinomial Logit Bandits.** There are two types of multinomial logit bandits studied in the literature: the single-parameter model, where the parameter is a vector [Cheung and Simchi-Levi, 2017] and multiple-parameter model, where the parameter is a matrix [Amani and Thrampoulidis, 2021]. We focus on the single-parameter model, which are more relevant to our setting. The pioneering work by Cheung and Simchi-Levi [2017] achieved a Bayesian regret of $\widetilde{\mathcal{O}}(\kappa^{-1}d\sqrt{T})$, where $T$ denotes the number of rounds in bandits. This result was further enhanced by subsequent studies [Oh and Iyengar, 2019, 2021, Agrawal et al., 2023]. In particular, Périvier and Goyal [2022] significantly improved the dependence on $\kappa$, obtaining a regret of $\widetilde{\mathcal{O}}(d\sqrt{\kappa T} + \kappa^{-1})$ in the uniform revenue setting. Most prior methods required storing all historical data and faced computational challenge. To address this issue, the most recent work by Lee and Oh [2024] proposed an algorithm with constant computational and storage costs building on recent advances in multiple-parameter model [Zhang and Sugiyama, 2023]. Their algorithm achieves the optimal regret of $\widetilde{\mathcal{O}}(d\sqrt{\kappa T} + \kappa^{-1})$ and $\widetilde{\mathcal{O}}(d\sqrt{T} + \kappa^{-1})$ under uniform and non-uniform rewards respectively. However, although the underlying models of MNL bandits and MDPs share similarities, the challenges they present differ substantially, and techniques developed for MNL bandits cannot be directly applied to MNL MDPs. For example, in MNL bandits, the objective is to select a series of assortments with *varying* sizes that maximize the expected revenue, whereas in MNL MDPs, the goal is to choose *one* action at each stage to maximize the cumulative reward. Thus, it is necessary to design new algorithms tailored for MDPs to address these unique challenges.

# 3 Problem Setup

In this section, we present the problem setup of RL with multinomial logit function approximation.

**Inhomogeneous, Episodic MDPs.** An inhomogeneous, episodic MDP instance can be denoted by a tuple $\mathcal{M} = (\mathcal{S}, \mathcal{A}, H, \{\mathbb{P}_h\}_{h=1}^H, \{r_h\}_{h=1}^H)$, where $\mathcal{S}$ is the state space, $\mathcal{A}$ is the action space, $H$ is the length of each episode, $\mathbb{P}_h : \mathcal{S} \times \mathcal{A} \times \mathcal{S} \to [0, 1]$ is the transition kernel with $\mathbb{P}_h(s' \mid s, a)$ is being the probability of transferring to state $s'$ from state $s$ and taking action $a$ at stage $h$, $r_h : \mathcal{S} \times \mathcal{A} \to [0, 1]$ is the deterministic reward function. A policy $\pi = \{\pi_h\}_{h=1}^H$ is a collection of mapping $\pi_h$, where each $\pi_h : \mathcal{S} \to \Delta(\mathcal{A})$ is a function maps a state $s$ to distributions over $\mathcal{A}$ at stage $h$. For any policy $\pi$ and $(s, a) \in \mathcal{S} \times \mathcal{A}$, we define the action-value function $Q_h^\pi$ and value function $V_h^\pi$ as follows:

$$Q_h^\pi(s, a) = \mathbb{E}\left[\sum_{h'=h}^H r_{h'}(s_{h'}, a_{h'}) \,\Big|\, s_h = s, a_h = a\right], \quad V_h^\pi(s) = \mathbb{E}_{a \sim \pi_h(\cdot \mid s)}[Q_h^\pi(s, a)],$$

where the expectation of $Q_h^\pi$ is taken over the randomness of the transition $\mathbb{P}$ and policy $\pi$. The optimal value function $V_h^*$ and action-value function $Q_h^*$ given by $V_h^*(s) = \sup_\pi V_h^\pi(s)$ and $Q_h^*(s, a) = \sup_\pi Q_h^\pi(s, a)$. For any function $V : \mathcal{S} \to \mathbb{R}$, we define $[\mathbb{P}_h V](s, a) = \mathbb{E}_{s' \sim \mathbb{P}_h(\cdot \mid s, a)} V(s')$.

**Learning Protocol.** In the online MDP setting, the learner interacts with the environment without the knowledge of the transition kernel $\{\mathbb{P}_h\}_{h=1}^H$. We assume the reward function $\{r_h\}_{h=1}^H$ is deterministic and known to the learner. The interaction proceeds in $K$ episodes. At the beginning of episode $k$, the learner chooses a policy $\pi_k = \{\pi_{k,h}\}_{h=1}^H$. At each stage $h \in [H]$, starting from the initial state $s_{k,1}$, the learner observes the state $s_{k,h}$, chooses an action $a_{k,h}$ sampled from $\pi_{k,h}(\cdot \mid s_{k,h})$, obtains reward $r_h(s_{k,h}, a_{k,h})$ and transits to the next state $s_{k,h+1} \sim \mathbb{P}_h(\cdot \mid s_{k,h}, a_{k,h})$ for $h \in [H]$. The episode ends when $s_{H+1}$ is reached. The goal of the learner is to minimize regret, defined as

$$\text{Reg}(K) = \sum_{k=1}^K V_1^*(s_{k,1}) - \sum_{k=1}^K V_1^{\pi_k}(s_{k,1}),$$

which is the difference between the cumulative reward of the optimal policy and the learner's policy.

**Multinomial Logit (MNL) Mixture MDPs.** Although significant advances have been achieved for MDPs with linear function approximation, Hwang and Oh [2023] show that there exists a set of features such that no linear transition model can induce a valid probability distribution, which limits the expressiveness of function approximation. To overcome this limitation, they propose a new class of MDPs with multinomial logit function approximation. However, their work focuses on the *homogeneous* setting, where the transitions remain the same across all stages (i.e., $\mathbb{P}_1 = ... = \mathbb{P}_H$). In this work, we address the more general *inhomogeneous* setting, allowing transitions to vary across different stages. We introduce the formal definition of inhomogeneous MNL mixture MDPs below.

**Definition 1** (Reachable States). For any $(h, s, a) \in [H] \times \mathcal{S} \times \mathcal{A}$, we define the "reachable states" as the set of states that can be reached from state $s$ taking action $a$ at stage $h$ within a single transition, i.e., $\mathcal{S}_{h,s,a} \triangleq \{s' \in \mathcal{S} \mid \mathbb{P}_h(s' \mid s, a) > 0\}$. Furthermore, we define $S_{h,s,a} \triangleq |\mathcal{S}_{h,s,a}|$ and denote by $U \triangleq \max_{(h,s,a)} S_{h,s,a}$ the maximum number of reachable states.

**Definition 2** (MNL Mixture MDP). An MDP instance $\mathcal{M} = (\mathcal{S}, \mathcal{A}, H, \{\mathbb{P}_h\}_{h=1}^H, \{r_h\}_{h=1}^H)$ is called an inhomogeneous, episodic $B$-bounded MNL mixture MDP if there exist a *known* feature mapping $\phi(s' \mid s, a) : \mathcal{S} \times \mathcal{A} \times \mathcal{S} \to \mathbb{R}^d$ with $\|\phi(s' \mid s, a)\|_2 \leq 1$ and *unknown* vectors $\{\theta_h^*\}_{h=1}^H \in \Theta$ with $\Theta = \{\theta \in \mathbb{R}^d, \|\theta\|_2 \leq B\}$, such that for all $(s, a, h) \in \mathcal{S} \times \mathcal{A} \times [H]$ and $s' \in \mathcal{S}_{h,s,a}$, it holds that

$$\mathbb{P}_h(s' \mid s, a) = \frac{\exp(\phi(s' \mid s, a)^\top \theta_h^*)}{\sum_{\widetilde{s} \in \mathcal{S}_{h,s,a}} \exp(\phi(\widetilde{s} \mid s, a)^\top \theta_h^*)}.$$

**Remark 1.** This model is consistent with models like neural networks [LeCun et al., 2015], where the feature $\phi$ is obtained by omitting the final layer, and $\theta_h^*$ represents the weights of the last layer. A final softmax layer is then applied to ensure that the output forms a valid probability distribution.

For any $\theta \in \mathbb{R}^d$, we define the induced transition as $p_{s,a}^{s'}(\theta) = \frac{\exp(\phi(s' \mid s, a)^\top \theta)}{\sum_{\widetilde{s} \in \mathcal{S}_{s,a}} \exp(\phi(\widetilde{s} \mid s, a)^\top \theta)}$. We then introduce the following two key problem-dependent quantities $\kappa$ and $\kappa^*$ that measure the effective non-linearity over the entire parameter space and around the ground truth parameter respectively.

**Assumption 1.** There exists $0 < \kappa \leq \kappa^* < 1$ such that for all $(s, a, h) \in \mathcal{S} \times \mathcal{A} \times [H]$ and $s', s'' \in \mathcal{S}_{h,s,a}$, it holds that $\inf_{\theta \in \Theta} p_{s,a}^{s'}(\theta) p_{s,a}^{s''}(\theta) \geq \kappa$ and $p_{s,a}^{s'}(\theta_h^*) p_{s,a}^{s''}(\theta_h^*) \geq \kappa^*$.

Assumption 1 is similar to the assumption in generalized linear bandit [Filippi et al., 2010] and logistic bandit [Faury et al., 2020, Abeille et al., 2021] to guarantee the Hessian matrix is non-singular.

Finally, we show the claim about the range of the magnitude of $\kappa$ and $\kappa^*$.

**Claim 1.** *It holds that* $1/(U \exp(2B))^2 \leq \kappa \leq \kappa^* \leq 1/U^2$.

# 4 Statistically Efficient Algorithm

The work of Hwang and Oh [2023] first introduced the MNL mixture MDPs and proposed an algorithm with a regret bound of $\widetilde{\mathcal{O}}(\kappa^{-1} d H^2 \sqrt{K})$. However, as discussed in Claim 1, it follows that $U^2 \leq \kappa^{-1} \leq (U \exp(2B))^2$, which results in the regret bound scaling polynomially with the number of reachable states $U$. In the worst case, $U$ can be equal to the size of the state space $S$, thereby undermining the motivation for function approximation, which aims to mitigate the dependence on the large state and action spaces. In this section, we address this significant issue by proposing a statistically efficient algorithm that eliminates this dependence in the dominant term of the regret.

## 4.1 Parameter Estimation

In this section, we first present the parameter estimation method based on the maximum likelihood estimation (MLE) for MNL mixture MDPs. Next, we review the confidence set construction based on the estimated parameters from previous work [Hwang and Oh, 2023]. Finally, we propose our new confidence set construction and highlight the improvements it offers over the previous approach.

Since the transition parameter $\theta_h^*$ is unknown, we need to estimate it using the historical data. At episode $k$, we collect a trajectory $\{(s_{k,h}, a_{k,h})\}_{h=1}^{H}$, then define the variable: $y_{k,h} \in \{0,1\}^{S_{k,h}}$ where $y_{k,h}^{s'} = \mathbb{1}_{\{s' = s_{k,h+1}\}}$ for $s' \in \mathcal{S}_{k,h} \triangleq \mathcal{S}_{s_{k,h}, a_{k,h}}$ and $S_{k,h} = |\mathcal{S}_{k,h}|$. We denote by $p_{k,h}^{s'}(\theta) = p_{s_{k,h}, a_{k,h}}^{s'}(\theta)$. Then $y_{k,h}$ is a sample from the following multinomial distribution:

$$y_{k,h} \sim \text{multinomial}(1, [p_{k,h}^{s_1}(\theta^*), \ldots, p_{k,h}^{s_{S_{k,h}}}(\theta^*)]),$$

where the parameter 1 indicates that $y_{k,h}$ is a single-trial sample. Furthermore, we define the noise $\epsilon_{k,h}^{s'} = y_{k,h}^{s'} - p_{k,h}^{s'}(\theta_h^*)$. It is clear that $\epsilon_{k,h} \in [-1,1]^{S_{k,h}}$, $\mathbb{E}[\epsilon_{k,h}] = \mathbf{0}$ and $\sum_{s' \in \mathcal{S}_{k,h}} \epsilon_{i,h}^{s'} = 0$.

We estimate the parameter $\theta_h^*$ using the MLE and construct the estimator $\widehat{\theta}_{k,h}$ as follows:

$$\widehat{\theta}_{k,h} = \underset{\theta \in \mathbb{R}^d}{\arg\min} \, \mathcal{L}_{k,h}(\theta) \triangleq \sum_{i=1}^{k-1} \sum_{s' \in \mathcal{S}_{i,h}} -y_{i,h}^{s'} \log p_{i,h}^{s'}(\theta) + \frac{\lambda_k}{2} \|\theta\|_2^2. \tag{1}$$

where $\lambda_k$ is the regularization parameter. Though the MLE estimator $\widehat{\theta}_{k,h}$ is the same as that of Hwang and Oh [2023], the confidence set is constructed differently. Specifically, define the gradient $\mathcal{G}_{k,h}(\theta)$ and Hessian matrix $\mathcal{H}_{k,h}(\theta)$ of the MLE loss by

$$\mathcal{G}_{k,h}(\theta) = \sum_{i=1}^{k-1} \sum_{s' \in \mathcal{S}_{i,h}} (p_{i,h}^{s'}(\theta) - y_{i,h}^{s'}) \phi_{i,h}^{s'} + \lambda_k \theta,$$

$$\mathcal{H}_{k,h}(\theta) = \sum_{i=1}^{k-1} \sum_{s' \in \mathcal{S}_{i,h}} p_{i,h}^{s'}(\theta) \phi_{i,h}^{s'}(\phi_{i,h}^{s'})^\top - \sum_{i=1}^{k-1} \sum_{s' \in \mathcal{S}_{i,h}} \sum_{s'' \in \mathcal{S}_{i,h}} p_{i,h}^{s'}(\theta) p_{i,h}^{s''}(\theta) \phi_{i,h}^{s'}(\phi_{i,h}^{s''})^\top + \lambda_k I.$$

Furthermore, we define the feature covariance matrix $A_{k,h} = \kappa^{-1} \lambda_k I + \sum_{i=1}^{k-1} \sum_{s' \in \mathcal{S}_{i,h}} \phi_{i,h}^{s'}(\phi_{i,h}^{s'})^\top$. By demonstrating $\mathcal{H}_{k,h}(\theta) \succeq \kappa A_{k,h}, \forall \theta \in \Theta$, Hwang and Oh [2023] construct the confidence set as

$$\mathcal{C}_{k,h} = \left\{ \left\| \theta - \widehat{\theta}_{k,h} \right\|_{A_{k,h}} \leq \kappa^{-1} \sqrt{d \log(kH/\delta)} \triangleq \beta_k \right\}. \tag{2}$$

Since the radius of the confidence set depends on $\kappa^{-1}$, the final regret bound also exhibits a dependence on $\kappa^{-1}$. To eliminates this dependence, we construct a $\kappa$-independent confidence set based on

new Bernstein-like inequalities in Lemma 13, following recent advances in logistic bandits [Faury et al., 2020, Périvier and Goyal, 2022]. Specifically, we show the following lemma.

**Lemma 1.** *For any $\delta \in (0,1)$, set $\lambda_k = d\log(kH/\delta)$ and define the confidence set as*

$$\widehat{\mathcal{C}}_{k,h} = \left\{\theta \in \Theta \mid \left\|\mathcal{G}_{k,h}(\theta) - \mathcal{G}_{k,h}(\widehat{\theta}_{k,h})\right\|_{\mathcal{H}_{k,h}^{-1}(\theta)} \le (B+3)\sqrt{d\log(kH/\delta)} \triangleq \widehat{\beta}_k\right\}. \quad (3)$$

*Then, we have $\Pr[\theta_h^* \in \widehat{\mathcal{C}}_{k,h}] \ge 1 - \delta, \forall k \in [K], h \in [H]$.*

**Comparison to prior work.** We compare the confidence sets defined in Eqs. (2) and (3).

For the confidence set in (3), by the self-concordance property of log-loss in Lemma 10, we have:

$$\left\|\theta - \widehat{\theta}_{k,h}\right\|_{\mathcal{H}_{k,h}(\theta)} \le (1 + 3\sqrt{2})\left\|\mathcal{G}_{k,h}(\theta) - \mathcal{G}_{k,h}(\widehat{\theta}_{k,h})\right\|_{\mathcal{H}_{k,h}^{-1}(\theta)} \le (1 + 3\sqrt{2})\widehat{\beta}_k.$$

Then, note that $\mathcal{H}_{k,h}(\theta) \succeq \kappa A_{k,h}$ for all $\theta \in \Theta$, we have

$$\left\|\theta - \widehat{\theta}_{k,h}\right\|_{A_{k,h}} \le \kappa^{-1/2}\left\|\theta - \widehat{\theta}_{k,h}\right\|_{\mathcal{H}_{k,h}(\theta)} \le \kappa^{-1/2}(1 + 3\sqrt{2})(B+3)\sqrt{d\log(kH/\delta)}. \quad (4)$$

Thus, compared to the confidence set in Eq. (2) from Hwang and Oh [2023], our confidence set in Eq. (3) provides a strict improvement by at least a factor of $\kappa^{-1/2}$. This improvement is one of the key components to eliminate the dependence on $\kappa^{-1}$ in the dominant term of the final regret bound.

Additionally, we identify a technical issue of Hwang and Oh [2023]. Specifically, they bound the confidence set in Eq. (2) using the self-normalized concentration in Lemma 12. However, the noise is not independent, and since $\sum_{s' \in \mathcal{S}_{i,h}} \epsilon_{i,h}^{s'} = 0$ (due to the learner visiting each stage $h$ exactly once per episode), it does not satisfy the *zero-mean* sub-Gaussian condition in Lemma 12. We observe similar oversights in multinomial logit contextual bandits [Oh and Iyengar, 2019, 2021, Agrawal et al., 2023], an issue that, to our knowledge, has not been explicitly addressed in prior work. This issue can be resolved with only slight modifications in constant factors by a new self-normalized concentration with dependent noises in Lemma 1 of Li et al. [2024b], a simplified version of Lemma 13.

## 4.2 Optimistic Value Function Construction

Given the confidence set $\widehat{\mathcal{C}}_{k,h}$, it is natural to follow the principle of "optimism in the face of uncertainty" and construct the optimistic value function. Hwang and Oh [2023] constructed the optimistic value function $\bar{Q}_{k,h}(s,a)$ by adding a closed-form upper confidence bound as follows:

$$\bar{Q}_{k,h}(s,a) = \left[r_h(s,a) + \sum_{s' \in \mathcal{S}_{h,s,a}} p_{s,a}^{s'}(\widehat{\theta}_{k,h})\bar{V}_{k,h+1}(s') + 2H\beta_k \max_{s' \in \mathcal{S}_{h,s,a}} \|\phi_{s,a}^{s'}\|_{A_{k,h}^{-1}}\right]_{[0,H]}, \quad (5)$$

where $\bar{V}_{k,h}(s) = \max_{a \in \mathcal{A}} \bar{Q}_{k,h}(s,a)$. Then, a naive idea to compute the optimistic value function is replacing the radius of the confidence set $\beta_k$ with $\widehat{\beta}_k$ and the feature covariance matrix $A_{k,h}$ with the Hessian matrix $\mathcal{H}_{k,h}(\theta_h^*)$. However, there are two issues with this approach. First, the true parameter $\theta_h^*$ is unknown thus the Hessian matrix $\mathcal{H}_{k,h}(\theta_h^*)$ is not computable in the algorithmic updates. Second, though $\widehat{\beta}_k$ is independent of $\kappa$ and the Hessian matrix $\mathcal{H}_{k,h}(\theta_h^*)$ captures local information, the bonus term $\max_{s' \in \mathcal{S}_{h,s,a}} \|\phi_{s,a}^{s'}\|_{\mathcal{H}_{k,h}^{-1}(\theta)}$ remains in a global form. This term involves taking the maximum over all states $s' \in \mathcal{S}_{h,s,a}$, which prevents fully utilizing the local information.

To address these challenges, we construct the optimistic value function by directly taking the maximum expected reward over the confidence set. Specifically, we define $\widehat{Q}_{k,h}(s,a)$ and $\widehat{V}_{k,h}(s)$ as

$$\widehat{Q}_{k,h}(s,a) = \left[r_h(s,a) + \max_{\theta \in \widehat{\mathcal{C}}_{k,h}} \sum_{s' \in \mathcal{S}_{h,s,a}} p_{s,a}^{s'}(\theta)\widehat{V}_{k,h+1}(s')\right]_{[0,H]}, \quad \widehat{V}_{k,h}(s) = \max_{a \in \mathcal{A}} \widehat{Q}_{k,h}(s,a). \quad (6)$$

This construction addresses the first challenge by eliminating the need for the Hessian matrix $\mathcal{H}_{k,h}(\theta_h^*)$ and directly leveraging the local information embedded in the confidence set $\widehat{\mathcal{C}}_{k,h}$. For the second challenge, although we bypass this issue in the construction of the optimistic value function, we still need to address it in the analysis. To tackle this, we employ a second-order Taylor expansion, in contrast to the first-order expansion used in the analysis of Hwang and Oh [2023]. This allows for a more precise capture of local information. Further details are provided in Lemma 7 in the appendix.

---

**Algorithm 1** UCRL-MNL-LL

---

**Input:** Regularization parameter $\lambda$, confidence width $\widehat{\beta}_k$, confidence parameter $\delta$.

1: **Initialization:** Set $\widehat{\theta}_{1,h} = \mathbf{0}, \widehat{Q}_{1,h}(\cdot,\cdot) = 0, \widehat{V}_{1,h}(\cdot) = 0$ for all $h \in [H]$.
2: **for** $k = 1, \ldots, K$ **do**
3:     **for** $h = 1, \ldots, H$ **do**
4:         Observe current state $s_{k,h}$ and select action $a_{k,h} = \arg\max_{a \in \mathcal{A}} \widehat{Q}_{k,h}(s_{k,h}, a)$.
5:     **end for**
6:     Set $\widehat{V}_{k+1,H+1}(\cdot) = 0$.
7:     **for** $h = H, \ldots, 1$ **do**
8:         Compute the estimator $\widehat{\theta}_{k+1,h}$ by Eq. (1) and update the confidence set $\widehat{\mathcal{C}}_{k+1,h}$ by Eq. (3).
9:         Compute $\widehat{Q}_{k+1,h}(\cdot,\cdot)$ and $\widehat{V}_{k+1,h}(\cdot)$ as in Eq. (6).
10:    **end for**
11: **end for**

---

### 4.3 Regret Guarantee

Based on the parameter estimation in Section 4.1 and the construction of the optimistic value function in Section 4.2, we propose the UCRL-MNL-LL algorithm. At each stage $h$ of episode $k$, the algorithm observes the current state $s_{k,h}$ and selects the action that maximizes the value function, i.e., $a_{k,h} = \arg\max_{a \in \mathcal{A}} \widehat{Q}_{k,h}(s_{k,h}, a)$, and transits to next state $s_{k,h+1}$. After collecting the trajectory $\{s_{k,h}, a_{k,h}\}_{h=1}^H$, the estimator $\widehat{\theta}_{k+1,h}$ is updated using Eq. (1), and the confidence set $\widehat{\mathcal{C}}_{k+1,h}$ is updated according to Eq. (3). Then, the value function $\widehat{Q}_{k+1,h}$ and $\widehat{V}_{k+1,h}$ are updated using Eq. (6). The detailed procedure is outlined in Algorithm 1. We show it achieves the following regret guarantee.

**Theorem 1.** *For any $\delta \in (0,1)$, set $\lambda_k = d\log(kH/\delta)$, and $\widehat{\beta}_k = (B+3)\sqrt{d\log(kH/\delta)}$. With probability at least $1 - \delta$, UCRL-MNL-LL algorithm (Algorithm 1) ensures the following guarantee:*

$$\mathrm{Reg}(K) \leq \widetilde{\mathcal{O}}\big(dH^2\sqrt{K} + \kappa^{-1}d^2H^2\big).$$

**Remark 2.** Focusing on the dominant term, our guarantee eliminates the problematic dependence on $\kappa^{-1}$, in stark contrast to the $\widetilde{\mathcal{O}}(\kappa^{-1}dH^2\sqrt{K})$ result of Hwang and Oh [2023]. As noted in Claim 1, such undesirable dependence has polynomial scaling with the number of reachable states $U$, which can be as large as the entire state space $S$ in the worst case. This renders the guarantee for function approximation—designed for settings with large state and action spaces—essentially vacuous.

## 5 Computationally Efficient Algorithm

While the UCRL-MNL-LL algorithm is the *first* statistically efficient algorithm for MNL mixture MDPs, it is computationally expensive dur the the optimization of the MLE in Eq. (1) and non-convex optimization in Eq. (6). To address these challenges, we propose a computationally efficient algorithm in this section, which attains the same regret but with constant computational costs per episode.

### 5.1 Efficient Online Parameter Estimation

In this section, we focus on estimating the unknown parameter $\theta_h^*$ in a computationally efficient manner. We first discuss the storage and time complexities of the MLE optimization in Eq. (1). Next, we introduce an efficient online parameter estimation based on online mirror descent that provides similar guarantees to the MLE, but with constant storage and time complexity per episode.

For the storage complexity, the optimization problem defined in Eq. (1) requires storing all historical data, resulting in a storage complexity of $\mathcal{O}(k)$ at episode $k$. In terms of time complexity, the problem does not have a closed-form solution and can only be solved to within an $\varepsilon$-accuracy, such as using projected gradient descent. As discussed in Faury et al. [2022], optimizing the MLE typically requires $\mathcal{O}(\log(1/\varepsilon))$ iterations to achieve an $\varepsilon$-accurate solution. Since the loss function is defined over all historical data, each gradient step incurs a query complexity of $\mathcal{O}(k)$. As $\varepsilon$ is usually chosen as $1/k$ for episode $k$, the total time complexity is $\mathcal{O}(k\log k)$ at episode $k$. Consequently, both storage and time complexities scale linearly with the episode count, which is computationally expensive.

To improve the computational efficiency, the basic idea is to estimate the unknown parameter with the online mirror descent (OMD) update instead of the MLE as defined in Eq. (1). To this end, we first define per-episode loss function $\ell_{k,h}(\theta)$, gradient $g_{k,h}(\theta)$ and Hessian matrix $H_{k,h}(\theta)$ as

$$\ell_{k,h}(\theta) = -\sum_{s' \in \mathcal{S}_{k,h}} y_{k,h}^{s'} \log p_{k,h}^{s'}(\theta), \quad g_{k,h}(\theta) = \nabla \ell_{k,h}(\theta) = \sum_{s' \in \mathcal{S}_{k,h}} (p_{k,h}^{s'}(\theta) - y_{k,h}^{s'})\phi_{k,h}^{s'} \quad (7)$$

$$H_{k,h}(\theta) = \nabla^2 \ell_{k,h}(\theta) = \sum_{s' \in \mathcal{S}_{k,h}} p_{k,h}^{s'}(\theta)\phi_{k,h}^{s'}(\phi_{k,h}^{s'})^\top - \sum_{s' \in \mathcal{S}_{k,h}} \sum_{s'' \in \mathcal{S}_{k,h}} p_{k,h}^{s'}(\theta)p_{k,h}^{s''}(\theta)\phi_{k,h}^{s'}(\phi_{k,h}^{s''})^\top.$$

Then, the design of the OMD algorithm can be conceptually divided into two parts: the approximation of the past losses and the approximation of the current loss. We provide the details of each below.

**Approximate the past losses.** To integrate historical information from previous iterations while avoiding the use of MLE in Eq. (1), we construct the estimator $\bar{\theta}_{k+1,h}$ using the implicit OMD form:

$$\bar{\theta}_{k+1,h} = \arg\min_{\theta \in \Theta} \left\{ \ell_{k,h}(\theta) + \frac{1}{2\eta} \|\theta - \bar{\theta}_{k,h}\|^2_{\bar{\mathcal{H}}_{k,h}} \right\}, \quad (8)$$

where $\eta$ is a step size and $\bar{\mathcal{H}}_{k,h} \triangleq \bar{\mathcal{H}}_{k,h}(\bar{\theta}_{k+1,h}) = \sum_{i=1}^{k-1} H_{i,h}(\bar{\theta}_{i+1,h}) + \lambda_k I$. The optimization problem can be decomposed in two terms. The first term is the instantaneous log-loss $\ell_{k,h}(\theta)$, which accounts for the information of the current episode. The second is a regularization term that ensures the current model remains close to the previous one, $\bar{\theta}_{k,h}$, thereby incorporating the historical information acquired so far. The most critical aspect in the above is the design of the local norm $\bar{\mathcal{H}}_{k,h}$, which intentionally approximate the per-episode Hessian matrix by $H_i(\bar{\theta}_{i+1,h})$ at a *look ahead* point $\bar{\theta}_{i+1,h}$. Such a Hessian matrix, originally introduced by Faury et al. [2022], effectively captures the local curvature of the loss function and is crucial for ensuring statistical efficiency.

The update rule in Eq. (8) is storage efficient, as it only requires storing the Hessian matrix $\bar{\mathcal{H}}_{k,h}$, which can be updated incrementally, resulting in an $\mathcal{O}(1)$ storage cost. In terms of time complexity, the optimization problem in Eq. (8) suffers an $\mathcal{O}(\log k)$ time complexity at episode $k$, since the loss function is defined only over the current episode. While this represents a significant improvement over the $\mathcal{O}(k \log k)$ time complexity of the MLE in Eq. (1), there is still a need to reduce the cost further to $\mathcal{O}(1)$ per episode, particularly given the potentially large number of episodes.

**Approximate the current loss.** To achieve $\mathcal{O}(1)$ time complexity per episode, we can further approximate the current loss with a second order approximation. Drawing inspiration from Zhang and Sugiyama [2023], we define the second-order approximation of the original loss function $\ell_{k,h}(\theta)$ at $\widetilde{\theta}_{k,h}$ as $\widetilde{\ell}_{k,h}(\theta) = \ell_{k,h}(\widetilde{\theta}_{k,h}) + \langle \nabla \ell_{k,h}(\widetilde{\theta}_{k,h}), \theta - \widetilde{\theta}_{k,h} \rangle + \frac{1}{2}\|\theta - \widetilde{\theta}_{k,h}\|^2_{H_{k,h}(\widetilde{\theta}_{k,h})}$, where $\widetilde{\theta}_{k,h}$ is the current estimate. Then, we can replace $\ell_{k,h}(\theta)$ with its second-order approximation $\widetilde{\ell}_{k,h}(\theta)$ in the optimization problem in Eq. (8). This leads to the following approximate optimization problem:

$$\widetilde{\theta}_{k+1,h} = \arg\min_{\theta \in \Theta} \left\{ \langle \nabla \ell_{k,h}(\widetilde{\theta}_{k,h}), \theta - \widetilde{\theta}_{k,h} \rangle + \frac{1}{2\eta} \|\theta - \widetilde{\theta}_{k,h}\|^2_{\widetilde{\mathcal{H}}_{k,h}} \right\}. \quad (9)$$

where $\eta$ is the step size, $\widetilde{\mathcal{H}}_{k,h} = \mathcal{H}_{k,h} + \eta H_{k,h}(\widetilde{\theta}_{k,h})$ and $\mathcal{H}_{k,h} = \sum_{i=1}^{k-1} H_{i,h}(\widetilde{\theta}_{i+1,h}) + \lambda_k I$. Then, Eq. (9) can be solved with a single projected gradient step with the following equivalent formulation:

$$\widetilde{\theta}'_{k+1,h} = \widetilde{\theta}_{k,h} - \eta \widetilde{\mathcal{H}}_{k,h}^{-1} \nabla \ell_{k,h}(\widetilde{\theta}_{k,h}), \quad \widetilde{\theta}_{k+1,h} = \arg\min_{\theta \in \Theta} \|\theta - \widetilde{\theta}'_{k+1,h}\|^2_{\widetilde{\mathcal{H}}_{k,h}}.$$

Thus, Eq. (9) is computationally efficient, as it only suffers an $\mathcal{O}(1)$ storage and time complexity.

Notice that the update rule in Eq. (9) is actually a standard online mirror descent (OMD) formula,

$$\widetilde{\theta}_{k+1,h} = \arg\min_{\theta \in \Theta} \left\{ \langle \nabla \ell_{k,h}(\widetilde{\theta}_{k,h}), \theta \rangle + \frac{1}{\eta} \mathcal{D}_{\psi_k}(\theta, \widetilde{\theta}_{k,h}) \right\}. \quad (10)$$

where the regularizer is $\psi_k(\theta) = \frac{1}{2}\|\theta\|^2_{\widetilde{\mathcal{H}}_{k,h}}$ and $\mathcal{D}_{\psi_k}(\cdot, \cdot)$ is the induced Bregman divergence. Therefore, we can construct the confidence set building upon the modern analysis of OMD [Orabona, 2019, Zhao et al., 2024]. Specifically, we can construct the $\kappa$-independent confidence set as follows.

**Lemma 2.** *For any $\delta \in (0, 1)$, set $\eta = \frac{1}{2}\log(1 + U) + (B + 1)$ and $\lambda = 84\sqrt{2}\eta(B + d)$, define*

$$\widetilde{\mathcal{C}}_{k,h} = \left\{ \theta \in \Theta \mid \|\theta - \widetilde{\theta}_{k,h}\|_{\mathcal{H}_{k,h}} \leq \widetilde{\beta}_k \right\},$$

*where $\widetilde{\beta}_k = \mathcal{O}(\sqrt{d}\log U \log(kH/\delta))$. Then, we have $\Pr[\theta_h^* \in \widetilde{\mathcal{C}}_{k,h}] \geq 1 - \delta, \forall k \in [K], h \in [H]$.*

**Remark 3.** Compared to the confidence set in Lemma 1, the radius $\widetilde{\beta}_k$ in Lemma 2 includes an additional $\log U$ factor. This is due to our approximation of the original MLE using the OMD update.

---

**Algorithm 2** UCRL-MNL-OL

---

**Input:** Step size $\eta$, regularization parameter $\lambda$, confidence width $\widetilde{\beta}_k$, confidence parameter $\delta$.

1: **Initialization:** $\mathcal{H}_{1,h} = \lambda I, \widehat{\theta}_{1,h} = \mathbf{0}$ for all $h \in [H]$.
2: **for** $k = 1, \ldots, K$ **do**
3:     Compute $\widetilde{Q}_{k,h}(\cdot, \cdot)$ in a backward way as in Eq. (11).
4:     **for** $h = 1, \ldots, H$ **do**
5:         Observe state $s_{k,h}$, select action $a_{k,h} = \arg\max_{a \in \mathcal{A}} \widetilde{Q}_{k,h}(s_{k,h}, a)$.
6:         Update $\widetilde{\mathcal{H}}_{k,h} = \mathcal{H}_{k,h} + \eta H_{k,h}(\widetilde{\theta}_{k,h})$.
7:         Compute $\widetilde{\theta}_{k+1,h} = \arg\min_{\theta \in \Theta} \langle \nabla \ell_{k,h}(\widetilde{\theta}_{k,h}), \theta - \widetilde{\theta}_{k,h} \rangle + \frac{1}{2\eta} \|\theta - \widetilde{\theta}_{k,h}\|^2_{\widetilde{\mathcal{H}}_{k,h}}$.
8:         Update $\mathcal{H}_{k+1,h} = \mathcal{H}_{k,h} + H_{k,h}(\widetilde{\theta}_{k+1,h})$.
9:     **end for**
10: **end for**

---

## 5.2 Efficient Optimistic Value Function Construction

Although the optimistic value function in Eq. (6) preserves local information effectively and provides strong theoretical guarantees, it is computationally intractable due to the need to solve a non-convex optimization problem. To address this challenge, we propose an efficient method in this section.

The key idea is to use a second-order Taylor expansion to derive a closed-form bonus term, which replaces the operation of taking the maximum over the non-convex confidence set. While this idea has been used in bandit settings, fundamental challenges arise when applying it in the MDP setting. Specifically, Zhang and Sugiyama [2023] studied the multi-parameter MLogB bandit, where each outcome is associated with a distinct parameter vector. In contrast, MNL mixture MDPs involve a single shared parameter vector across all outcomes. This distinction leads to a more complex Hessian matrix, necessitating a more sophisticated analysis. A direct use of their analysis will leads to a polynomial dependence on the number of reachable states $U$, which is undesirable in the MDP setting. Lee and Oh [2024] focused on the single-parameter MNL bandit, which is more closely related to our setting. However, they construct the optimistic value function by directly taking the maximum over the confidence set, a computationally intractable approach in the MDP setting. As a result, they can apply a second-order Taylor expansion around the ground truth parameter $\theta_h^*$ in their analysis, while we must apply it around the estimated parameter $\widetilde{\theta}_{k,h}$ to construct the bonus term explicitly.

For MDPs, we show the value difference arising from the transition estimation error as follows.

**Lemma 3.** *Suppose Lemma 2 holds. For any $V : \mathcal{S} \to [0, H]$ and $(h, s, a) \in [H] \times \mathcal{S} \times \mathcal{A}$, it holds*

$$\left| \sum_{s' \in \mathcal{S}_{h,s,a}} p_{s,a}^{s'}(\widetilde{\theta}_{k,h})V(s') - \sum_{s' \in \mathcal{S}_{h,s,a}} p_{s,a}^{s'}(\theta_h^*)V(s') \right| \leq \epsilon_{s,a}^{\mathtt{fst}} + \epsilon_{s,a}^{\mathtt{snd}}.$$

*where*

$$\epsilon_{s,a}^{\mathtt{fst}} = H\widetilde{\beta}_k \sum_{s' \in \mathcal{S}_{h,s,a}} p_{s,a}^{s'}(\widetilde{\theta}_{k,h}) \left\| \phi_{s,a}^{s'} - \sum_{s'' \in \mathcal{S}_{h,s,a}} p_{s,a}^{s''}(\widetilde{\theta}_{k,h})\phi_{s,a}^{s''} \right\|_{\mathcal{H}_{k,h}^{-1}}, \epsilon_{s,a}^{\mathtt{snd}} = \frac{5}{2}H\widetilde{\beta}_k^2 \max_{s' \in \mathcal{S}_{h,s,a}} \|\phi_{s,a}^{s'}\|^2_{\mathcal{H}_{k,h}^{-1}}.$$

Based on Lemma 3, we construct the optimistic value function as follows:

$$\widetilde{Q}_{k,h}(s,a) = \left[ r_h(s,a) + \sum_{s' \in \mathcal{S}_{h,s,a}} p_{s,a}^{s'}(\widetilde{\theta}_{k,h})\widetilde{V}_{k,h+1}(s') + \epsilon_{s,a}^{\mathtt{fst}} + \epsilon_{s,a}^{\mathtt{snd}} \right]_{[0,H]}, \tag{11}$$

where $\widetilde{V}_{k,h}(s) = \max_{a \in \mathcal{A}} \widetilde{Q}_{k,h}(s,a)$. In contrast to the value function in Eq. (5), which incorporates the term $\max_{s' \in \mathcal{S}_{h,s,a}} \|\phi_{s,a}^{s'}\|_{\mathcal{H}_{k,h}^{-1}}$, the refined value function in Eq. (11) replaces it with $\epsilon_{s,a}^{\mathtt{fst}} + \epsilon_{s,a}^{\mathtt{snd}}$. This modification better preserves local information, offering a more accurate estimation error bound.

## 5.3 Regret Guarantee

The overall algorithm UCRL-MNL-OL is similar to UCRL-MNL-LL, but with the estimator and optimistic value function updated in a computationally efficient manner. The detailed algorithm is presented in Algorithm 2. We provide the guarantee of UCRL-MNL-OL in the following theorem.

**Theorem 2.** *For any $\delta \in (0, 1)$, set $\widetilde{\beta}_k = \mathcal{O}(\sqrt{d}\log U \log(kH/\delta))$, $\eta = \frac{1}{2}\log(1 + U) + (B + 1)$ and $\lambda = 84\sqrt{2}\eta(B + d)$, with probability at least $1 - \delta$, UCRL-MNL-LL (Algorithm 2) ensures*

$$\text{Reg}(K) \leq \widetilde{\mathcal{O}}\big(dH^2\sqrt{K} + \kappa^{-1}d^2H^2\big).$$

**Remark 4.** UCRL-MNL-OL attains the same regret as UCRL-MNL-LL, but with constant computational cost per episode. This is achieved by constructing an efficient online estimation based on OMD and an optimistic value function by closed-form bonus instead of the non-convex optimization.

## 6  Lower Bound

In this section, we establish the lower bound for MNL mixture MDPs by presenting a novel reduction, which connects MNL mixture MDPs and the logistic bandit problem.

Consider the following logistic bandit problem [Faury et al., 2020]: at each round $t \in [T]$, the learner selects an action $x_t \in \mathcal{X}$ and receives a reward $r_t$ sampled from Bernoulli distribution with mean $\mu(x^\top\theta^*) = (1 + \exp(-x^\top\theta^*))^{-1}$, where $\theta^* \in \{\theta \in \mathbb{R}^d, \|\theta\|_2 \leq B\}$ is the unknown parameter. The learner aims to to minimize the regret: $\text{Reg}^{\text{LogB}}(T) = \max_{x \in \mathcal{X}} \sum_{t=1}^T \mu(x^\top\theta^*) - \sum_{t=1}^T \mu(x_t^\top\theta^*)$.

**Theorem 3.** *For any logistic bandit problem $\mathcal{B}$, there exists an MNL mixture MDP $\mathcal{M}$ such that learning $\mathcal{M}$ is as hard as learning $H/2$ independent instances of $\mathcal{B}$ simultaneously.*

**Corollary 1** (Lower Bound). *For any problem instance $\{\theta_h^*\}_{h=1}^H$ and for $K \geq d^2\kappa^*$, there exists an MNL mixture MDP with* infinite *action space such that $\text{Reg}(K) \geq \Omega(dH\sqrt{K\kappa^*})$.*

**Remark 5.** Corollary 1 also implies a problem-independent lower bound of $\Omega(dH\sqrt{K})$ directly. Corollary 1 can be proved by combining Theorem 3 and the $\Omega(d\sqrt{T\kappa^*})$ lower bound for logistic bandits with *infinite* arms by Abeille et al. [2021]. To the best of our knowledge, a lower bound for logistic bandits with finite arms has not been established, which is beyond the scope of this work. This absence leaves the lower bound for MNL mixture MDPs with a finite action space open through this reduction. However, after the submission of our work to arXiv [Li et al., 2024a], a follow up work by Park et al. [2024] proposed a new reduction that bridges MNL mixture MDPs with linear mixture MDPs by approximating MNL functions to linear functions. Leveraging this new reduction, they established a problem-independent $\Omega(dH^{3/2}\sqrt{K})$ lower bound for the finite action setting. This achievement confirms that our result is optimal in $d$ and $K$, only loosing by an $\mathcal{O}(H^{1/2})$ factor.

**Dependence on $H$.** By the discussion in Remark 5, we note that our result is optimal with respect to $d$ and $K$, but loosing by an $\mathcal{O}(H^{1/2})$ factor. We discuss the challenges in improving the dependence on $H$. Notably, MNL mixture MDPs can be viewed as a generalization of linear mixture MDPs [Ayoub et al., 2020, Zhou et al., 2021]. The pioneering work by Ayoub et al. [2020] achieved a regret bound of $\widetilde{\mathcal{O}}(dH^2\sqrt{K})$ for linear mixture MDPs, which matches our results in Theorem 1, differing only on the lower-order term. Later, Zhou et al. [2021] enhanced the dependence on $H$ and attained an optimal regret bound of $\widetilde{\mathcal{O}}(d\sqrt{H^3K})$. This was made possible by recognizing that the value function in linear mixture MDPs is linear, allowing for direct learning of the value function while incorporating *variance information*. In contrast, the value function for MNL mixture MDPs does not conform to a specific structure, posing a significant challenge in using the variance information of value functions. Thus, it remains open whether similar improvements on $H$ are attainable for MNL mixture MDPs.

## 7  Conclusion and Future Work

In this work, we addressing both statistical and computational challenges for MNL mixture MDPs, which leverage MNL function approximation to ensure valid probability distributions. Specifically, we propose a statistically efficient algorithm that achieve a regret of $\widetilde{\mathcal{O}}(dH^2\sqrt{K} + \kappa^{-1}d^2H^2)$, eliminating the dependence on $\kappa^{-1}$ in the dominant term for the first time. Then, we introduce a computationally enhanced algorithm that achieves the same regret but with only constant cost. Finally, we establish the first lower bound for this problem, justifying the optimality of our results in $d$ and $K$.

There are several interesting directions for future work. First, there still exists a gap between the upper and lower bounds. How to close this gap remains an open problem. Besides, we focuses on stationary rewards in this work, extending MNL mixture MDPs to the non-stationary settings and studying the dynamic regret [Wei and Luo, 2021, Zhao et al., 2022, Li et al., 2023] is also an important direction.

## Acknowledgments

This research was supported by National Science and Technology Major Project (2022ZD0114800) and NSFC (U23A20382, 62206125). Peng Zhao was supported in part by the Xiaomi Foundation.

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

# A  Notations

In this section, we collect the notations used in the paper in Table 2.

Table 2: Notations used in the regret analysis.

| Notation | Definition and description |
|---|---|
| $\ell_{k,h}(\theta)$ | $\triangleq -\sum_{s'\in\mathcal{S}_{k,h}} y_{k,h}^{s'} \log p_{k,h}^{s'}(\theta)$, per-episode loss function at episode $k$ and stage $h$ |
| $g_{k,h}(\theta)$ | $\triangleq \nabla\ell_{k,h}(\theta) = \sum_{s'\in\mathcal{S}_{k,h}} (p_{k,h}^{s'}(\theta) - y_{k,h}^{s'})\phi_{k,h}^{s'}$, gradient of loss $\ell_{k,h}(\theta)$ |
| $H_{k,h}(\theta)$ | $\triangleq \sum_{s'\in\mathcal{S}_{k,h}} p_{k,h}^{s'}(\theta)\phi_{k,h}^{s'}(\phi_{k,h}^{s'})^\top - \sum_{s',\in\mathcal{S}_{k,h}}\sum_{s''\in\mathcal{S}_{k,h}} p_{k,h}^{s'}(\theta)p_{k,h}^{s''}(\theta)\phi_{k,h}^{s'}(\phi_{k,h}^{s''})^\top$ |
| $\mathcal{L}_{k,h}(\theta)$ | $\triangleq \sum_{i=1}^{k-1}\ell_{i,h}(\theta) + \frac{\lambda_k}{2}\|\theta\|_2^2$, the cumulative MLE loss |
| $\mathcal{G}_{k,h}(\theta)$ | $\triangleq \nabla\mathcal{L}_{k,h}(\theta) = \sum_{i=1}^{k-1}\sum_{s'\in\mathcal{S}_{i,h}} (p_{i,h}^{s'}(\theta)-y_{i,h}^{s'})\phi_{i,h}^{s'} + \lambda_k\theta$, gradient of $\mathcal{L}_{k,h}(\theta)$ |
| $\mathcal{H}_{k,h}(\theta)$ | $\triangleq \nabla^2\mathcal{L}_{k,h}(\theta) = \sum_{i=1}^{k-1} H_{i,h}(\theta) + \lambda_k I_d$, Hessian of MLE loss $\mathcal{L}_{k,h}(\theta)$ |
| $\widehat{\theta}_{k,h}$ | $\triangleq \arg\min_{\theta\in\mathbb{R}^d} \mathcal{L}_{k,h}(\theta)$, the MLE estimator at episode $k$ and stage $h$ |
| $\mathcal{H}_{k,h}$ | $\triangleq \mathcal{H}_{k,h}(\widetilde{\theta}_{i+1,h}) = \sum_{i=1}^{k-1} H_{i,h}(\widetilde{\theta}_{i+1,h}) + \lambda_k I_d$, the cumulative *look ahead* Hessian |
| $\widetilde{\mathcal{H}}_{k,h}$ | $\triangleq \mathcal{H}_{k,h} + \eta H_{k,h}(\widetilde{\theta}_{k,h})$, the sum of *look ahead* Hessian and the Hessian of current loss |
| $\widetilde{\theta}_{k+1,h}$ | $\triangleq \arg\min_{\theta\in\Theta}\langle\nabla\ell_{k,h}(\widetilde{\theta}_{k,h}),\theta-\widetilde{\theta}_{k,h}\rangle + \frac{1}{2\eta}\|\theta-\widetilde{\theta}_{k,h}\|_{\widetilde{\mathcal{H}}_{k,h}}^2$, the OMD estimator |

# B  Properties of Multinomial Logit Function

This section collects several key properties of the multinomial logit function used in the paper.

Without loss of generality, we assume $\forall\mathcal{S}_{h,s,a}, \exists\dot{s}_{h,s,a}\in\mathcal{S}_{h,s,a}$ such that $\phi(\dot{s}\mid s,a)=\mathbf{0}$. Otherwise, we can always define a new feature mapping $\phi'(s''\mid s,a) = \phi(s'\mid s,a) - \phi(s''\mid s,a)$ for any $s''\in\mathcal{S}_{h,s,a}$ such that $\phi'(s'\mid s,a)=\mathbf{0}$ and the transition kernel induced by $\phi'$ is the same as that induced by $\phi$. Furthermore, We denote the set $\dot{\mathcal{S}}_{h,s,a} = \mathcal{S}_{h,s,a}\backslash\{\dot{s}_{h,s,a}\}$.

First, we introduce the definition of self-concordant-like functions and demonstrate that the MNL loss function is self-concordant-like.

**Definition 3** (Self-concordant-like function, Tran-Dinh et al. [2015]). *A convex function $f\in\mathcal{C}^3(\mathbb{R}^m)$ is $M$-self-concordant-like function with constant $M$ if:*
$$|\psi'''(s)| \leqslant M\|\mathbf{b}\|_2\psi''(s).$$
*for $s\in\mathbb{R}$ and $M>0$, where $\psi(s) := f(\mathbf{a}+s\mathbf{b})$ for any $\mathbf{a},\mathbf{b}\in\mathbb{R}^m$.*

**Proposition 1.** *The per-episode MNL loss $\ell_{k,h}(\theta)$ and the cumulative MNL loss $\mathcal{L}_{k,h}(\theta)$ are both $3\sqrt{2}$-self-concordant-like for all $k\in[K], h\in[H]$.*

*Proof.* By proposition B.1 in Lee and Oh [2024], the per-episode MNL loss function $\ell_{k,h}(\theta)$ is $3\sqrt{2}$-self-concordant-like. Then, the cumulative MNL loss function $\mathcal{L}_{k,h}(\theta)$ is the sum of self-concordant-like functions and a quadratic function, it is also $3\sqrt{2}$-self-concordant-like. $\blacksquare$

**Lemma 4** (Zhang and Sugiyama [2023, Lemma 1]). *Let $\ell(\mathbf{z},y) = \sum_{k=0}^{K}\mathbf{1}\{y=k\}\cdot\log\left(\frac{1}{[\sigma(\mathbf{z})]_k}\right)$ where $\sigma(\mathbf{z})_k = \frac{e^{z_k}}{\sum_{j=0}^{K} e^{z_j}}$, $\mathbf{a}\in[-C,C]^K$, $y\in\{0\}\cup[K]$ and $\mathbf{b}\in\mathbb{R}^K$ where $C>0$. Then, we have*
$$\ell(\mathbf{a},y) \geq \ell(\mathbf{b},y) + \nabla\ell(\mathbf{b},y)^\top(\mathbf{a}-\mathbf{b}) + \frac{1}{\log(K+1)+2(C+1)}(\mathbf{a}-\mathbf{b})^\top\nabla^2\ell(\mathbf{b},y)(\mathbf{a}-\mathbf{b}).$$

Then, we show the Hessian of the MNL loss function is positive semi-definite.

**Lemma 5.** *The following statements hold for any $k \in [K], h \in [H]$:*

$$H_{k,h}(\theta) \succeq \sum_{s' \in \dot{\mathcal{S}}_{k,h}} p_{k,h}^{s'}(\theta) p_{k,h}^{\dot{s}_{k,h}}(\theta) \phi_{k,h}^{s'}(\phi_{k,h}^{s'})^\top \succeq \kappa \sum_{s' \in \dot{\mathcal{S}}_{k,h}} \phi_{k,h}^{s'}(\phi_{k,h}^{s'})^\top.$$

*Proof.* First, note that

$$\forall x, y \in \mathbb{R}^d, (x - y)(x - y)^\top = xx^\top + yy^\top - xy^\top - yx^\top \succeq 0 \implies xx^\top + yy^\top \succeq xy^\top + yx^\top.$$

Then, we have

$$
\begin{aligned}
H_{k,h}(\theta) &= \sum_{s' \in \mathcal{S}_{k,h}} p_{k,h}^{s'}(\theta) \phi_{k,h}^{s'}(\phi_{k,h}^{s'})^\top - \sum_{s' \in \mathcal{S}_{k,h}} \sum_{s'' \in \mathcal{S}_{k,h}} p_{k,h}^{s'}(\theta) p_{k,h}^{s''}(\theta) \phi_{k,h}^{s'}(\phi_{k,h}^{s''})^\top \\
&= \sum_{s' \in \dot{\mathcal{S}}_{k,h}} p_{k,h}^{s'}(\theta) \phi_{k,h}^{s'}(\phi_{k,h}^{s'})^\top - \frac{1}{2} \sum_{s' \in \dot{\mathcal{S}}_{k,h}} \sum_{s'' \in \dot{\mathcal{S}}_{k,h}} p_{k,h}^{s'}(\theta) p_{k,h}^{s''}(\theta) \left( \phi_{k,h}^{s'}(\phi_{k,h}^{s''})^\top + \phi_{k,h}^{s''}(\phi_{k,h}^{s'})^\top \right) \\
&\succeq \sum_{s' \in \dot{\mathcal{S}}_{k,h}} p_{k,h}^{s'}(\theta) \phi_{k,h}^{s'}(\phi_{k,h}^{s'})^\top - \frac{1}{2} \sum_{s' \in \dot{\mathcal{S}}_{k,h}} \sum_{s'' \in \dot{\mathcal{S}}_{k,h}} p_{k,h}^{s'}(\theta) p_{k,h}^{s''}(\theta) \left( \phi_{k,h}^{s'}(\phi_{k,h}^{s'})^\top + \phi_{k,h}^{s''}(\phi_{k,h}^{s''})^\top \right) \\
&= \sum_{s' \in \dot{\mathcal{S}}_{k,h}} p_{k,h}^{s'}(\theta) \phi_{k,h}^{s'}(\phi_{k,h}^{s'})^\top - \sum_{s' \in \dot{\mathcal{S}}_{k,h}} \sum_{s'' \in \dot{\mathcal{S}}_{k,h}} p_{k,h}^{s'}(\theta) p_{k,h}^{s''}(\theta) \phi_{k,h}^{s'}(\phi_{k,h}^{s'})^\top \\
&= \sum_{s' \in \dot{\mathcal{S}}_{k,h}} p_{k,h}^{s'}(\theta) \left( 1 - \sum_{s'' \in \dot{\mathcal{S}}_{k,h}} p_{k,h}^{s''}(\theta) \right) \phi_{k,h}^{s'}(\phi_{k,h}^{s'})^\top \\
&= \sum_{s' \in \dot{\mathcal{S}}_{k,h}} p_{k,h}^{s'}(\theta) p_{k,h}^{\dot{s}_{k,h}}(\theta) \phi_{k,h}^{s'}(\phi_{k,h}^{s'})^\top \\
&\succeq \kappa \sum_{s' \in \dot{\mathcal{S}}_{k,h}} \phi_{k,h}^{s'}(\phi_{k,h}^{s'})^\top,
\end{aligned}
$$

where the last inequality holds by the definition of $\kappa$ in Assumption 1. This finishes the proof. ∎

Next, we show several concentration inequalities commonly used in the analysis.

**Lemma 6.** *Suppose $\lambda_k \geq 1$, for any $k \in [K], h \in [H]$, for the quantities in Table 2 and define*

$$\bar{\phi}_{k,h}^{s'} = \phi_{k,h}^{s'} - \sum_{s'' \in \mathcal{S}_{k,h}} p_{k,h}^{s''}(\theta_h^*) \phi_{k,h}^{s''}, \quad \widetilde{\phi}_{k,h}^{s'} = \phi_{k,h}^{s'} - \sum_{s'' \in \mathcal{S}_{k,h}} p_{k,h}^{s''}(\widetilde{\theta}_{k+1,h}) \phi_{k,h}^{s''}.$$

*Then, the following statements hold:*

(I) $\displaystyle \sum_{i=1}^{k} \sum_{s' \in \mathcal{S}_{i,h}} p_{i,h}^{s'}(\theta_h^*) \|\bar{\phi}_{i,h}^{s'}\|_{\mathcal{H}_{i,h}^{-1}(\theta_h^*)}^2 \leq 2d \log \left( 1 + \frac{k}{\lambda_k d} \right)$

(II) $\displaystyle \sum_{i=1}^{k} \sum_{s' \in \mathcal{S}_{i,h}} p_{i,h}^{s'}(\widetilde{\theta}_{i+1,h}) \|\widetilde{\phi}_{i,h}^{s'}\|_{\mathcal{H}_{i,h}^{-1}}^2 \leq 2d \log \left( 1 + \frac{k}{\lambda_k d} \right)$

(III) $\displaystyle \sum_{i=1}^{k} \sum_{s' \in \mathcal{S}_{i,h}} p_{i,h}^{s'}(\widetilde{\theta}_{i+1,h}) p_{i,h}^{\dot{s}_{k,h}}(\widetilde{\theta}_{i+1,h}) \|\phi_{i,h}^{s'}\|_{\mathcal{H}_{i,h}^{-1}}^2 \leq 2d \log \left( 1 + \frac{k}{\lambda_k d} \right)$

(IV) $\displaystyle \sum_{i=1}^{k} \max_{s' \in \mathcal{S}_{i,h}} \|\phi_{i,h}^{s'}\|_{\mathcal{H}_{i,h}^{-1}(\theta)}^2 \leq \frac{2}{\kappa} d \log \left( 1 + \frac{k}{\lambda_k d} \right), \forall \theta \in \Theta$

(V) $\displaystyle \sum_{i=1}^{k} \max_{s' \in \mathcal{S}_{i,h}} \|\widetilde{\phi}_{i,h}^{s'}\|_{\mathcal{H}_{i,h}^{-1}}^2 \leq \frac{2}{\kappa} d \log \left( 1 + \frac{k}{\lambda_k d} \right).$

*Proof.* We prove the five statements individually.

**Proof of statement (I).** By the definition of $H_{k,h}(\theta)$, we have

$$H_{i,h}(\theta) = \sum_{s' \in \mathcal{S}_{i,h}} p_{i,h}^{s'}(\theta) \phi_{i,h}^{s'}(\phi_{i,h}^{s'})^\top - \sum_{s' \in \mathcal{S}_{i,h}} \sum_{s'' \in \mathcal{S}_{i,h}} p_{i,h}^{s'}(\theta) p_{i,h}^{s''}(\theta) \phi_{i,h}^{s'}(\phi_{i,h}^{s''})^\top$$

$$= \mathbb{E}_{s' \in p_{i,h}(\theta)}[\phi_{i,h}^{s'}(\phi_{i,h}^{s'})^\top] - \mathbb{E}_{s' \in p_{i,h}(\theta)}[\phi_{i,h}^{s'}]\big(\mathbb{E}_{s'' \in p_{i,h}(\theta)}[\phi_{i,h}^{s''}]\big)^\top$$

$$= \mathbb{E}_{s' \in p_{i,h}(\theta)}\big[(\phi_{i,h}^{s'} - \mathbb{E}_{s'' \in p_{i,h}(\theta)}\phi_{i,h}^{s''})(\phi_{i,h}^{s'} - \mathbb{E}_{s'' \in p_{i,h}(\theta)}\phi_{i,h}^{s''})^\top\big] \tag{12}$$

Thus, we have $H_{i,h}(\theta_h^*) \succeq \sum_{s' \in \mathcal{S}_{i,h}} p_{i,h}(\theta_h^*)(\bar{\phi}_{i,h}^{s'})(\bar{\phi}_{i,h}^{s'})^\top$. Then, we get

$$\mathcal{H}_{i+1,h}(\theta_h^*) \succeq \mathcal{H}_{i,h}(\theta_h^*) + \sum_{s' \in \mathcal{S}_{i,h}} p_{i,h}(\theta_h^*)(\bar{\phi}_{i,h}^{s'})(\bar{\phi}_{i,h}^{s'})^\top$$

As a result, we have

$$\det(\mathcal{H}_{i+1,h}(\theta_h^*)) \geq \det(\mathcal{H}_{i,h}(\theta_h^*))\Big(1 + \sum_{s' \in \mathcal{S}_{i,h}} p_{i,h}(\theta_h^*)\|\bar{\phi}_{i,h}^{s'}\|_{\mathcal{H}_{i,h}^{-1}(\theta_h^*)}^2\Big).$$

Since $\lambda \geq 1$, we have $\sum_{s' \in \mathcal{S}_{i,h}} p_{i,h}(\theta_h^*)\|\bar{\phi}_{i,h}^{s'}\|_{\mathcal{H}_{i,h}^{-1}(\theta_h^*)}^2 \leq 1$. Using the fact that $z \leq 2\log(1+z)$ for any $z \in [0,1]$, we get

$$\sum_{i=1}^k \sum_{s' \in \mathcal{S}_{i,h}} p_{i,h}(\theta_h^*)\|\bar{\phi}_{i,h}^{s'}\|_{\mathcal{H}_{i,h}^{-1}(\theta_h^*)}^2 \leq 2 \sum_{i=1}^k \log\Big(1 + \sum_{s' \in \mathcal{S}_{i,h}} p_{i,h}(\theta_h^*)\|\bar{\phi}_{i,h}^{s'}\|_{\mathcal{H}_{i,h}^{-1}(\theta_h^*)}^2\Big)$$

$$\leq 2\log\Big(\frac{\det(\mathcal{H}_{k+1,h}(\theta_h^*))}{\det(\mathcal{H}_{1,h}(\theta_h^*))}\Big)$$

$$\leq 2d\log\Big(1 + \frac{k}{\lambda d}\Big),$$

where the last inequality holds by the determinant inequality in Lemma 14.

**Proof of statement (II).** The proof is same as that of (I), except that we replace $\theta_h^*$ with $\widetilde{\theta}_{i+1,h}$.

**Proof of statement (III).** By Lemma 5, we have $H_{k,h}(\theta) \succeq \sum_{s' \in \dot{\mathcal{S}}_{k,h}} p_{k,h}^{s'}(\theta) p_{k,h}^{\dot{s}_{k,h}}(\theta) \phi_{k,h}^{s'}(\phi_{k,h}^{s'})^\top$. The remaining proof is the same as the proof of statement (I).

**Proof of statement (IV).** By Lemma 5, we have $\forall \theta \in \Theta$, it holds that $\mathcal{H}_{k+1,h}(\theta) \succeq \mathcal{H}_{k,h}(\theta) + \kappa \sum_{s' \in \dot{\mathcal{S}}_{k,h}} \phi_{k,h}^{s'}(\phi_{k,h}^{s'})^\top$. Since $\lambda \geq 1$, we have $\kappa \max_{s' \in \mathcal{S}_{i,h}} \|\phi_{i,h}^{s'}\|_{\mathcal{H}_{i,h}^{-1}(\theta)} \leq \kappa$. Using the fact that $z \leq 2\log(1+z)$ for any $z \in [0,1]$. By a similar analysis as the statement (I), we have

$$\sum_{i=1}^k \max_{s' \in \mathcal{S}_{i,h}} \|\phi_{i,h}^{s'}\|_{\mathcal{H}_{i,h}^{-1}}^2 \leq \frac{2}{\kappa} \sum_{i=1}^k \log\Big(1 + \kappa \max_{s' \in \mathcal{S}_{i,h}} \|\phi_{i,h}^{s'}\|_{\mathcal{H}_{i,h}^{-1}}\Big)$$

$$\leq \frac{2}{\kappa} \sum_{i=1}^k \log\Big(1 + \kappa \sum_{s' \in \mathcal{S}_{i,h}} \|\phi_{i,h}^{s'}\|_{\mathcal{H}_{i,h}^{-1}}\Big)$$

$$\leq \frac{2}{\kappa} \log\Big(\frac{\det(\mathcal{H}_{k+1,h}(\theta))}{\det(\mathcal{H}_{1,h}(\theta))}\Big)$$

$$\leq \frac{2}{\kappa} d\log\Big(1 + \frac{k}{\lambda d}\Big).$$

This finishes the proof of statement (IV).

**Proof of statement (V).** By (12), we have

$$H_{i,h}(\widetilde{\theta}_{i+1,h}) \succeq \sum_{s' \in \mathcal{S}_{i,h}} p_{i,h}(\widetilde{\theta}_{i+1,h})(\widetilde{\phi}_{i,h}^{s'})(\widetilde{\phi}_{i,h}^{s'})^\top \succeq \kappa \sum_{s' \in \mathcal{S}_{i,h}} (\widetilde{\phi}_{i,h}^{s'})(\widetilde{\phi}_{i,h}^{s'})^\top.$$

Thus, we have

$$\mathcal{H}_{k+1,h} \succeq \mathcal{H}_{k,h} + \kappa \sum_{s' \in \dot{\mathcal{S}}_{k,h}} \widetilde{\phi}_{k,h}^{s'}(\widetilde{\phi}_{k,h}^{s'})^\top.$$

Then, the remaining proof is similar to the proof of statement (III). ∎

# C    Useful Lemmas for MNL Mixture MDPs

In this section, we present some useful lemmas that are commonly used in the analysis.

## C.1    Useful Lemmas

**Lemma 7.** *For any $\theta_1, \theta_2 \in \mathbb{R}^d$ and positive semi-definite matrix $\Lambda$, suppose $\|\theta_1 - \theta_2\|_\Lambda \leq \beta$. Then, for any $V : \mathcal{S} \to [0, H]$ and $(h, s, a) \in [H] \times \mathcal{S} \times \mathcal{A}$, it holds*

$$\left| \sum_{s' \in \mathcal{S}_{h,s,a}} p_{s,a}^{s'}(\theta_1) V(s') - \sum_{s' \in \mathcal{S}_{h,s,a}} p_{s,a}^{s'}(\theta_2) V(s') \right| \leq \epsilon_{s,a}^{\texttt{1st}} + \epsilon_{s,a}^{\texttt{2nd}}.$$

*where*

$$\epsilon_{s,a}^{\texttt{1st}} = H\beta \sum_{s' \in \mathcal{S}_{h,s,a}} p_{s,a}^{s'}(\theta_1) \left\| \phi_{s,a}^{s'} - \sum_{s'' \in \mathcal{S}_{h,s,a}} p_{s,a}^{s''}(\theta_1) \phi_{s,a}^{s''} \right\|_{\Lambda^{-1}},$$

$$\epsilon_{s,a}^{\texttt{2nd}} = \frac{5}{2} H\beta^2 \max_{s'} \|\phi_{s,a}^{s'}\|_{\Lambda^{-1}}^2.$$

**Lemma 8.** *Suppose $\forall (k, h, s, a) \in K \times [H] \times \mathcal{S} \times \mathcal{A}$ and $\widehat{\theta}_{k,h} \in \mathbb{R}^d$, it holds that $\theta_h^* \in \widehat{\mathcal{C}}_{k,h}$ where*

$$\widehat{\mathcal{C}}_{k,h} = \left\{ \theta \,\Big|\, \left| \sum_{s' \in \mathcal{S}_{h,s,a}} p_{s,a}^{s'}(\widehat{\theta}_{k,h}) V(s') - \sum_{s' \in \mathcal{S}_{h,s,a}} p_{s,a}^{s'}(\theta) V(s') \right| \leq \Gamma_{k,h,s,a} \right\}. \tag{13}$$

*Define*

$$\widehat{Q}_{k,h}(s,a) = \left[ r_h(s,a) + \arg\max_{\theta \in \mathcal{C}_{k,h}} \sum_{s' \in \mathcal{S}_{h,s,a}} p_{s,a}^{s'}(\theta) \widehat{V}_{k,h+1}(s') \right]_{[0,H]}, \tag{14}$$

*or,*

$$\widehat{Q}_{k,h}(s,a) = \left[ r_h(s,a) + \sum_{s' \in \mathcal{S}_{h,s,a}} p_{s,a}^{s'}(\widehat{\theta}_{k,h}) \widehat{V}_{k,h+1}(s') + \Gamma_{k,h,s,a} \right]_{[0,H]}, \tag{15}$$

*where $\widehat{V}_{k,h}(s) = \max_{a \in \mathcal{A}} \widehat{Q}_{k,h}(s,a)$. Select the action as $a_{k,h} = \arg\max_{a \in \mathcal{A}} \widehat{Q}_{k,h}(s_{k,h}, a)$. Then, for any $\delta \in (0, 1]$, then it holds that*

$$Q_h^*(s,a) \leq \widehat{Q}_{k,h}(s,a) \leq r_h(s,a) + \mathbb{P}_h \widehat{V}_{k,h+1}(s,a) + 2\Gamma_{k,h,s,a}.$$

**Lemma 9.** *Suppose Lemma 8 holds. Then, it holds that*

$$\text{Reg}(K) \leq 2 \sum_{k=1}^{K} \sum_{h=1}^{H} \Gamma_{k,h,s_{k,h},a_{k,h}} + H\sqrt{2KH\log(2/\delta)}.$$

## C.2    Proof of Lemma 7

*Proof.* By the second-order Taylor expansion at $\theta_1$, there exists $\bar{\theta} = \nu\theta_1 + (1-\nu)\theta_2$ for some $\nu \in [0, 1]$, such that

$$\sum_{s' \in \mathcal{S}_{h,s,a}} p_{s,a}^{s'}(\theta_2) V(s') - \sum_{s' \in \mathcal{S}_{h,s,a}} p_{s,a}^{s'}(\theta_1) V(s')$$

$$= \sum_{s' \in \mathcal{S}_{h,s,a}} \nabla p_{s,a}^{s'}(\theta_1)^\top (\theta_2 - \theta_1) V(s') + \frac{1}{2} \sum_{s' \in \mathcal{S}_{h,s,a}} (\theta_2 - \theta_1)^\top \nabla^2 p_{s,a}^{s'}(\bar{\theta}) (\theta_2 - \theta_1) V(s')$$

The gradient of $p_{s,a}^{s'}(\theta)$ is given by

$$\nabla p_{s,a}^{s'}(\theta) = p_{s,a}^{s'}(\theta) \phi_{s,a}^{s'} - p_{s,a}^{s'}(\theta) \sum_{s'' \in \mathcal{S}_{h,s,a}} p_{s,a}^{s''}(\theta) \phi_{s,a}^{s''}.$$

For the first-order term, we have

$$\sum_{s'\in\mathcal{S}_{h,s,a}} \nabla p^{s'}_{s,a}(\theta_1)^\top(\theta_2-\theta_1)V(s')$$

$$= \sum_{s'\in\mathcal{S}_{h,s,a}} p^{s'}_{s,a}(\theta_1)(\phi^{s'}_{s,a})^\top(\theta_2-\theta_1)V(s') - \sum_{s'\in\mathcal{S}_{h,s,a}} p^{s'}_{s,a}(\theta_1)\sum_{s''\in\mathcal{S}_{h,s,a}} p^{s''}_{s,a}(\theta_1)(\phi^{s''}_{s,a})^\top(\theta_2-\theta_1)V(s')$$

$$\le H \sum_{s'\in\mathcal{S}^+_{h,s,a}} p^{s'}_{s,a}(\theta_1)\left((\phi^{s'}_{s,a})^\top(\theta_2-\theta_1) - \sum_{s''\in\mathcal{S}_{h,s,a}} p^{s''}_{s,a}(\theta_1)(\phi^{s''}_{s,a})^\top(\theta_2-\theta_1)\right)$$

$$= H \sum_{s'\in\mathcal{S}^+_{h,s,a}} p^{s'}_{s,a}(\theta_1)\left(\left(\phi^{s'}_{s,a} - \sum_{s''\in\mathcal{S}_{h,s,a}} p^{s''}_{s,a}(\theta_1)\phi^{s''}_{s,a}\right)^\top(\theta_2-\theta_1)\right)$$

$$\le H \sum_{s'\in\mathcal{S}^+_{h,s,a}} p^{s'}_{s,a}(\theta_1)\left(\left\|\phi^{s'}_{s,a} - \sum_{s''\in\mathcal{S}_{h,s,a}} p^{s''}_{s,a}(\theta_1)\phi^{s''}_{s,a}\right\|_{\Lambda^{-1}}\|\theta_2-\theta_1\|_\Lambda\right)$$

$$\le H\beta \sum_{s'\in\mathcal{S}^+_{h,s,a}} p^{s'}_{s,a}(\theta_1)\left\|\phi^{s'}_{s,a} - \sum_{s''\in\mathcal{S}_{h,s,a}} p^{s''}_{s,a}(\theta_1)\phi^{s''}_{s,a}\right\|_{\Lambda^{-1}}$$

$$\le H\beta \sum_{s'\in\mathcal{S}_{h,s,a}} p^{s'}_{s,a}(\theta_1)\left\|\phi^{s'}_{s,a} - \sum_{s''\in\mathcal{S}_{h,s,a}} p^{s''}_{s,a}(\theta_1)\phi^{s''}_{s,a}\right\|_{\Lambda^{-1}} \tag{16}$$

where in the first inequality, we denote $\mathcal{S}^+_{h,s,a}$ as the subset of $\mathcal{S}_{h,s,a}$ such that $(\phi^{s'}_{s,a})^\top(\theta_2-\theta_1) - \sum_{s''\in\mathcal{S}_{h,s,a}} p^{s''}_{s,a}(\theta_2)(\phi^{s''}_{s,a})^\top(\theta_2-\theta_1)$ is non-negative, the second inequality holds by the Holder's inequality, and the third inequality is by the condition $\|\theta_1-\theta_2\|_\Lambda \le \beta$.

For the second-order term, let $u^{s'}_{s,a}(\theta) = (\phi^{s'}_{s,a})^\top\theta$ and $p^{s'}_{s,a}(u) = \frac{\exp(u^{s'}_{s,a})}{1+\sum_{s''}\exp(u^{s''}_{s,a})}$, further define

$$F(u) = \sum_{s'\in\mathcal{S}_{h,s,a}} \frac{\exp(u^{s'}_{s,a})}{1+\sum_{s''\in\mathcal{S}_{h,s,a}}\exp(u^{s''}_{s,a})}, \quad \widetilde{F}(u) = \sum_{s'\in\mathcal{S}_{h,s,a}} \frac{\exp(u^{s'}_{s,a})V(s')}{1+\sum_{s''\in\mathcal{S}_{h,s,a}}\exp(u^{s''}_{s,a})}.$$

Then, we have

$$\frac{1}{2}\sum_{s'\in\mathcal{S}_{h,s,a}} (\theta_2-\theta_1)^\top\nabla^2 p^{s'}_{s,a}(\bar\theta)(\theta_2-\theta_1)V(s')$$

$$= \frac{1}{2}\big(u(\theta_2)-u(\theta_1)\big)^\top\nabla^2\widetilde{F}(u(\bar\theta))\big(u(\theta_2)-u(\theta_1)\big)$$

$$= \frac{1}{2}\sum_{s'\in\mathcal{S}_{h,s,a}}\sum_{s''\in\mathcal{S}_{h,s,a}} \big(u^{s'}_{s,a}(\theta_2)-u^{s'}_{s,a}(\theta_1)\big)^\top\frac{\partial^2\widetilde{F}(u(\bar\theta))}{\partial s'\partial s''}\big(u^{s''}_{s,a}(\theta_2)-u^{s''}_{s,a}(\theta_1)\big)$$

$$\le \frac{H}{2}\sum_{s'\in\mathcal{S}_{h,s,a}}\sum_{s''\in\mathcal{S}_{h,s,a}} \big|u^{s'}_{s,a}(\theta_2)-u^{s'}_{s,a}(\theta_1)\big|\cdot\frac{\partial^2 F(u(\bar\theta))}{\partial s'\partial s''}\cdot\big|u^{s''}_{s,a}(\theta_2)-u^{s''}_{s,a}(\theta_1)\big|$$

where the inequality holds by $V(s)\in[0,H],\forall s$.

According to Lemma 17, we have (omit the subscript $\mathcal{S}_{h,s,a}$ for simplicity):

$$\frac{H}{2}\sum_{s'}\sum_{s''}\big|u^{s'}_{s,a}(\theta_2)-u^{s'}_{s,a}(\theta_1)\big|\cdot\frac{\partial^2 F(u(\bar\theta))}{\partial s'\partial s''}\cdot\big|u^{s''}_{s,a}(\theta_2)-u^{s''}_{s,a}(\theta_1)\big|$$

$$\le H\sum_{s'}\sum_{s''\ne s'} \big|u^{s'}_{s,a}(\theta_2)-u^{s'}_{s,a}(\theta_1)\big|\cdot p^{s'}_{s,a}(u(\bar\theta))p^{s''}_{s,a}(u(\bar\theta))\cdot\big|u^{s''}_{s,a}(\theta_2)-u^{s''}_{s,a}(\theta_1)\big|$$

$$+ \frac{3H}{2}\sum_{s'}\big(u^{s'}_{s,a}(\theta_2)-u^{s'}_{s,a}(\theta_1)\big)^2 p^{s'}_{s,a}(u(\bar\theta)). \tag{17}$$

To bound the first term, by applying the AM-GM inequality, we obtain

$$
H \sum_{s'} \sum_{s'' \neq s'} \left| u_{s,a}^{s'}(\theta_2) - u_{s,a}^{s'}(\theta_1) \right| \cdot p_{s,a}^{s'}\big(u(\bar{\theta})\big) p_{s,a}^{s''}\big(u(\bar{\theta})\big) \cdot \left| u_{s,a}^{s''}(\theta_2) - u_{s,a}^{s''}(\theta_1) \right|
$$

$$
\leq H \sum_{s'} \sum_{s''} \left| u_{s,a}^{s'}(\theta_2) - u_{s,a}^{s'}(\theta_1) \right| \cdot p_{s,a}^{s'}\big(u(\bar{\theta})\big) p_{s,a}^{s''}\big(u(\bar{\theta})\big) \cdot \left| u_{s,a}^{s''}(\theta_2) - u_{s,a}^{s''}(\theta_1) \right|
$$

$$
\leq \frac{H}{2} \sum_{s'} \sum_{s''} \left( u_{s,a}^{s'}(\theta_2) - u_{s,a}^{s'}(\theta_1) \right)^2 p_{s,a}^{s'}\big(u(\bar{\theta})\big) p_{s,a}^{s''}\big(u(\bar{\theta})\big)
$$

$$
+ \frac{H}{2} \sum_{s'} \sum_{s''} \left( u_{s,a}^{s''}(\theta_2) - u_{s,a}^{s''}(\theta_1) \right)^2 p_{s,a}^{s'}\big(u(\bar{\theta})\big) p_{s,a}^{s''}\big(u(\bar{\theta})\big)
$$

$$
\leq H \sum_{s'} \left( u_{s,a}^{s'}(\theta_2) - u_{s,a}^{s'}(\theta_1) \right)^2 p_{s,a}^{s'}\big(u(\bar{\theta})\big) \tag{18}
$$

Plugging (18) into (17), we have

$$
\frac{H}{2} \sum_{s'} \sum_{s''} \left| u_{s,a}^{s'}(\theta_2) - u_{s,a}^{s'}(\theta_1) \right| \cdot \frac{\partial^2 F(u(\bar{\theta}))}{\partial s' \partial s''} \cdot \left| u_{s,a}^{s''}(\theta_2) - u_{s,a}^{s''}(\theta_1) \right|
$$

$$
\leq \frac{5H}{2} \sum_{s'} \left( u_{s,a}^{s'}(\theta_2) - u_{s,a}^{s'}(\theta_1) \right)^2 p_{s,a}^{s'}(u(\bar{\theta}))
$$

$$
= \frac{5H}{2} \sum_{s'} \left( (\phi_{s,a}^{s'})^\top (\theta_2 - \theta_1) \right)^2 p_{s,a}^{s'}(u(\bar{\theta}))
$$

$$
\leq \frac{5H}{2} \beta^2 \max_{s'} \| \phi_{s,a}^{s'} \|_{\Lambda^{-1}}^2, \tag{19}
$$

where the last inequality holds by the condition $\| \theta_1 - \theta_2 \|_\Lambda \leq \beta$.

Finally, combining (16) and (19) finishes the proof. ∎

### C.3 Proof of Lemma 8

*Proof.* First, we prove the left-hand side of the lemma. We prove this by backward induction on $h$. For the stage $h = H$, by definition, we have $\widehat{Q}_{k,H}(s,a) = r_H(s,a) = Q_H^*(s,a), \widehat{V}_{k,H+1}(s) = 0 = V_{H+1}^*(s)$. Suppose the statement holds for $h + 1$, we show it holds for $h$. By definition, if $\widehat{Q}_{k,h}(s,a) = H$, this holds trivially. Otherwise, we consider two cases:

For $\widehat{Q}_{k,h}(s,a)$ defined in (14), we have

$$
\widehat{Q}_{k,h}(s,a) = r_h(s,a) + \arg\max_{\theta \in \mathcal{C}_{k,h}} \sum_{s' \in \mathcal{S}_{h,s,a}} p_{s,a}^{s'}(\theta) \widehat{V}_{k,h+1}(s')
$$

$$
\geq r_h(s,a) + \arg\max_{\theta \in \mathcal{C}_{k,h}} \sum_{s' \in \mathcal{S}_{h,s,a}} p_{s,a}^{s'}(\theta) V_{k,h+1}^*(s')
$$

$$
\geq r_h(s,a) + \sum_{s' \in \mathcal{S}_{h,s,a}} p_{s,a}^{s'}(\theta_h^*) V_{k,h+1}^*(s') = Q_h^*(s,a).
$$

where the first inequality is by the induction hypothesis, and the second inequality is due to $\theta_h^* \in \mathcal{C}_{k,h}$.

For $\widehat{Q}_{k,h}(s,a)$ defined in (15), we have

$$
\widehat{Q}_{k,h}(s,a) = r_h(s,a) + \sum_{s' \in \mathcal{S}_{h,s,a}} p_{s,a}(\widehat{\theta}_{k,h}) \widehat{V}_{k,h+1}(s') + \Gamma_{h,s,a}
$$

$$
\geq r_h(s,a) + \sum_{s' \in \mathcal{S}_{h,s,a}} p_{s,a}(\widehat{\theta}_{k,h}) V_{k,h+1}^*(s') + \Gamma_{h,s,a}
$$

$$
\geq r_h(s,a) + \sum_{s' \in \mathcal{S}_{h,s,a}} p_{s,a}(\theta_h^*) V_{k,h+1}^*(s') = Q_h^*(s,a).
$$

where the first inequality is by the induction hypothesis, and the second inequality is by (13).

Then, we prove the right-hand side of the lemma.

For $\widehat{Q}_{k,h}(s,a)$ defined in (14), we have

$$
\begin{aligned}
\widehat{Q}_{k,h}(s,a) &= r_h(s,a) + \arg\max_{\theta \in \mathcal{C}_{k,h}} \sum_{s' \in \mathcal{S}_{h,s,a}} p_{s,a}^{s'}(\theta)\widehat{V}_{k,h+1}(s') \\
&\leq r_h(s,a) + \sum_{s' \in \mathcal{S}_{h,s,a}} p_{s,a}^{s'}(\widehat{\theta}_{k,h})\widehat{V}_{k,h+1}(s') + \Gamma_{k,h,s,a} \\
&\leq r_h(s,a) + \sum_{s' \in \mathcal{S}_{h,s,a}} p_{s,a}^{s'}(\theta_h^*)\widehat{V}_{k,h+1}(s') + 2\Gamma_{h,s,a},
\end{aligned}
$$

where the inequality is by (13).

For $\widehat{Q}_{k,h}(s,a)$ defined in (15), we have

$$
\begin{aligned}
\widehat{Q}_{k,h}(s,a) &= r_h(s,a) + \sum_{s' \in \mathcal{S}_{h,s,a}} p_{s,a}^{s'}(\widehat{\theta}_{k,h})\widehat{V}_{k,h+1}(s') + \Gamma_{k,h,s,a} \\
&\leq r_h(s,a) + \sum_{s' \in \mathcal{S}_{h,s,a}} p_{s,a}^{s'}(\theta_h^*)\widehat{V}_{k,h+1}(s') + 2\Gamma_{k,h,s,a},
\end{aligned}
$$

where the inequality is by (13). ∎

### C.4  Proof of Lemma 9

*Proof.* By the definition that $\mathrm{Reg}(K) = \sum_{k=1}^{K} V_1^*(s_{k,1}) - \sum_{k=1}^{K} V_1^{\pi_k}(s_{k,1})$, we have for any $\delta \in (0,1]$, with probability at least $1 - \delta$, it holds that

$$
\begin{aligned}
\sum_{k=1}^{K} V_1^*(s_{k,1}) - \sum_{k=1}^{K} V_1^{\pi_k}(s_{k,1}) &= \sum_{k=1}^{K} Q_1^*(s_{k,1}, \pi^*(s_{k,1})) - \sum_{k=1}^{K} V_1^{\pi_k}(s_{k,1}) \\
&\leq \sum_{k=1}^{K} \widehat{Q}_1(s_{k,1}, \pi^*(s_{k,1})) - \sum_{k=1}^{K} V_1^{\pi_k}(s_{k,1}) \\
&\leq \sum_{k=1}^{K} \widehat{Q}_1(s_{k,1}, a_{k,1}) - \sum_{k=1}^{K} V_1^{\pi_k}(s_{k,1}),
\end{aligned}
$$

where the first inequality is by Lemma 8 and $\theta_h^* \in \widehat{\mathcal{C}}_{k,h}$ with probability at least $1 - \delta$, and the second inequality is by action selection $a_{k,h} = \arg\max_{a \in \mathcal{A}} \widehat{Q}_{k,h}(s_{k,h}, a)$.

By the right-hand side of Lemma 8, we have

$$
\begin{aligned}
&\widehat{Q}_1(s_{k,1}, a_{k,1}) - V_1^{\pi_k}(s_{k,1}) \\
&= r(s_{k,1}, a_{k,1}) + \mathbb{P}_1 \widehat{V}_{k,2}(s_{k,1}, a_{k,1}) + 2\Gamma_{h,s_{k,1},a_{k,1}} - r(s_{k,1}, a_{k,1}) - \mathbb{P}_1 V_2^{\pi_k}(s_{k,1}, a_{k,1}) \\
&\leq \mathbb{P}_1(\widehat{V}_{k,2} - V_2^{\pi_k})(s_{k,1}, a_{k,1}) - (\widehat{V}_{k,2} - V_2^{\pi_k})(s_{k,2}) + (\widehat{V}_{k,2} - V_2^{\pi_k})(s_{k,2}) + 2\Gamma_{h,s_{k,1},a_{k,1}} \\
&\leq \mathbb{P}_1(\widehat{V}_{k,2} - V_2^{\pi_k})(s_{k,1}, a_{k,1}) - (\widehat{V}_{k,2} - V_2^{\pi_k})(s_{k,2}) + \left(\widehat{Q}_2(s_{k,2}, a_{k,2}) - V_2^{\pi_k}(s_{k,2})\right) + 2\Gamma_{h,s_{k,1},a_{k,1}}.
\end{aligned}
$$

Define $\mathcal{M}_{k,h} = \mathbb{P}_h(\widehat{V}_{k,h+1} - V_{h+1}^{\pi_k})(s_{k,h}, a_{k,h}) - (\widehat{V}_{k,h+1} - V_{h+1}^{\pi_k})(s_{k,h+1})$. Applying this recursively, we have

$$
\widehat{Q}_1(s_{k,1}, a_{k,1}) - V_1^{\pi_k}(s_{k,1}) \leq 2 \sum_{h=1}^{H} \Gamma_{h,s_{k,h},a_{k,h}} + \sum_{h=1}^{H} \mathcal{M}_{k,h}
$$

Summing over $k$, we have

$$
\mathrm{Reg}(K) \leq 2 \sum_{k=1}^{K} \sum_{h=1}^{H} \Gamma_{h,s_{k,h},a_{k,h}} + \sum_{k=1}^{K} \sum_{h=1}^{H} \mathcal{M}_{k,h} \leq 2 \sum_{k=1}^{K} \sum_{h=1}^{H} \Gamma_{h,s_{k,h},a_{k,h}} + H\sqrt{2KH\log(2/\delta)}
$$

where the inequality holds by the Azuma-Hoeffding inequality as $\mathcal{M}_{k,h}$ is a martingale difference sequence with $\mathcal{M}_{k,h} \leq 2H$. This finishes the proof. ∎

# D  Omitted Proofs for Section 3

## D.1  Proof of Claim 1

*Proof.* First, by the definition of MNL mixture MDP, we have $\forall (s,a) \in \mathcal{S} \times \mathcal{A}$ and $s' \in \mathcal{S}_{h,s,a}$, it holds that $p_{s,a}^{s'}(\theta) \geq \exp(-B)/(U\exp(B)), \forall \theta \in \mathbb{R}^d$, thus $\kappa^* \geq \kappa \geq 1/(U\exp(2B))^2$. Next, consider the state-action pair $(s,a)$ at stage $h$ with the maximum number of reachable states $U$, it is clear that $\sum_{s' \in \mathcal{S}_{h,s,a}} p_{s,a}^{s'}(\theta_h^*) = 1$. This implies that $\sum_{s' \in \mathcal{S}_{h,s,a}} \sum_{s'' \in \mathcal{S}_{h,s,a}} p_{s,a}^{s'}(\theta_h^*) p_{s,a}^{s''}(\theta_h^*) = 1$. Applying the pigeonhole principle, there exists $s', s'' \in \mathcal{S}_{h,s,a}$ such that $p_{s,a}^{s'}(\theta_h^*) p_{s,a}^{s''}(\theta_h^*) \leq 1/U^2$. Thus, we conclude that $\kappa \leq \kappa^* \leq 1/U^2$. This finishes the proof. ∎

# E  Omitted Proofs for Section 4

## E.1  Useful Lemma

**Lemma 10.** $\forall \theta_1, \theta_2 \in \Theta$, *we have* $\|\theta_1 - \theta_2\|_{\mathcal{H}_{k,h}(\theta_1)} \leq (1 + 3\sqrt{2})\|\mathcal{G}_{k,h}(\theta_1) - \mathcal{G}_{k,h}(\theta_2)\|_{\mathcal{H}_{k,h}^{-1}(\theta_1)}$.

*Proof.* By the multivariate mean value theorem, we have

$$\mathcal{G}_{k,h}(\theta_1) - \mathcal{G}_{k,h}(\theta_2) = \nabla\mathcal{L}_{k,h}(\theta_1) - \nabla\mathcal{L}_{k,h}(\theta_2) = \int_0^1 \nabla^2 \mathcal{L}_{k,h}(\theta_2 + t(\theta_1 - \theta_2))dt(\theta_1 - \theta_2).$$

Hence, we have

$$\|\mathcal{G}(\theta_1) - \mathcal{G}(\theta_2)\|_{G_{k,h}^{-1}(\theta_1,\theta_2)} = \|\theta_1 - \theta_2\|_{G_{k,h}(\theta_1,\theta_2)}.$$

where $G_{k,h}(\theta_1, \theta_2) = \int_0^1 \nabla^2 \mathcal{L}_{k,h}(\theta_2 + t(\theta_1 - \theta_2))dt$. By self-concordant-like property of $\mathcal{L}_{k,h}$ in Proposition 1, we have $\mathcal{H}_{k,h}(\theta_1) \preceq (1 + 3\sqrt{2})G_{k,h}(\theta_1, \theta_2)$. As a result, we have

$$\begin{aligned}
\|\theta_1 - \theta_2\|_{\mathcal{H}_{k,h}(\theta_1)} &\leq (1 + 3\sqrt{2})^{1/2} \|\theta_1 - \theta_2\|_{G_{k,h}(\theta_1,\theta_2)} \\
&= (1 + 3\sqrt{2})^{1/2} \|\mathcal{G}_{k,h}(\theta_1) - \mathcal{G}_{k,h}(\theta_2)\|_{G_{k,h}^{-1}(\theta_1,\theta_2)} \\
&\leq (1 + 3\sqrt{2}) \|\mathcal{G}_{k,h}(\theta_1) - \mathcal{G}_{k,h}(\theta_2)\|_{\mathcal{H}_{k,h}^{-1}(\theta_1)}.
\end{aligned}$$

This finishes the proof. ∎

## E.2  Proof of Lemma 1

*Proof.* Since $\widehat{\theta}_{k,h}$ minimizes $\mathcal{L}_{k,h}(\theta)$, we have $\mathcal{G}_{k,h}(\widehat{\theta}_{k,h}) = \mathbf{0}$. Thus, we have

$$\mathcal{G}_{k,h}(\theta_h^*) - \mathcal{G}_{k,h}(\widehat{\theta}_{k,h}) = \sum_{i=1}^{k-1} \sum_{s' \in \mathcal{S}_{i,h}} (p_{i,h}^{s'}(\theta_h^*) - y_{i,h}^{s'})\phi_{i,h}^{s'} + \lambda_k \theta_h^*.$$

Therefore, since $\|\theta_h^*\| \leq B$ and $\mathcal{H}_{k,h}(\theta_h^*) \succeq \lambda_k I$, for any $\delta \in (0,1]$, with probability at least $1 - \frac{\delta}{H}$,

$$\begin{aligned}
\|\mathcal{G}_{k,h}(\theta_h^*) - \mathcal{G}_{k,h}(\widehat{\theta}_{k,h})\|_{\mathcal{H}_{k,h}^{-1}(\theta_h^*)} &\leq \left\|\sum_{i=1}^{k-1} \sum_{s' \in \mathcal{S}_{i,h}} (p_{i,h}^{s'}(\theta_h^*) - y_{i,h}^{s'})\phi_{i,h}^{s'}\right\|_{\mathcal{H}_{k,h}^{-1}(\theta_h^*)} + \sqrt{\lambda_k}B \\
&\leq \frac{\sqrt{\lambda_k}}{4} + \frac{4}{\sqrt{\lambda_k}} \log\left(\frac{2^d H \det(\mathcal{H}_{k,h}(\theta_h^*))^{\frac{1}{2}} \lambda_k^{-\frac{d}{2}}}{\delta}\right) + \sqrt{\lambda_k}B,
\end{aligned}$$

where the last inequality holds by the Bernstein-type concentration inequality in Lemma 13. Then, by the determinant inequality in Lemma 14, we have $\det(\mathcal{H}_{k,h}(\theta_h^*)) \leq \left(\lambda_k + \frac{k}{d}\right)^d$. Thus, we obtain

$$\|\mathcal{G}_{k,h}(\theta_h^*) - \mathcal{G}_{k,h}(\widehat{\theta}_{k,h})\|_{\mathcal{H}_{k,h}^{-1}(\theta_h^*)} \leq \left(B + \frac{1}{2}\right)\sqrt{\lambda_k} + \frac{2}{\sqrt{\lambda_k}} \log\left(\frac{4}{\delta}\left(1 + \frac{kH}{d\lambda_k}\right)\right).$$

By the configuration that $\lambda_k = d\log(kH/\delta)$ and applying the union bound for $h \in [H]$, we have with probability at least $1 - \delta$, for all $h \in [H]$ simultaneously, it holds that

$$\|\mathcal{G}_{k,h}(\theta_h^*) - \mathcal{G}_{k,h}(\widehat{\theta}_{k,h})\|_{\mathcal{H}_{k,h}^{-1}(\theta_h^*)} \leq (B + 3)\sqrt{d\log(kH/\delta)}.$$

This finishes the proof. ∎

### E.3 Proof of Theorem 1

*Proof.* By Lemma 10, with probability at least $1 - \delta$, it holds that

$$\|\widehat{\theta}_{k,h} - \theta_h^*\|_{\mathcal{H}_{k,h}^{-1}(\theta_h^*)} \leq (1 + 3\sqrt{2})\|\mathcal{G}_{k,h}(\widehat{\theta}_{k,h}) - \mathcal{G}_{k,h}(\theta_h^*)\|_{\mathcal{H}_{k,h}^{-1}(\theta_h^*)} \leq (1 + 3\sqrt{2})\widehat{\beta}_k.$$

where the last inequality holds the confidence set $\widehat{\mathcal{C}}_{k,h}$ in Lemma 1. Then, by Lemma 7, we have

$$\left| \sum_{s' \in \mathcal{S}_{h,s,a}} p_{s,a}^{s'}(\widehat{\theta}_{k,h})V(s') - \sum_{s' \in \mathcal{S}_{h,s,a}} p_{s,a}^{s'}(\theta_h^*)V(s') \right| \leq \epsilon_{s,a}^{\mathtt{1st}} + \epsilon_{s,a}^{\mathtt{2nd}}.$$

where

$$\epsilon_{s,a}^{\mathtt{1st}} = (1 + 3\sqrt{2})H\widehat{\beta}_k \sum_{s' \in \mathcal{S}_{h,s,a}} p_{s,a}^{s'}(\theta_h^*)\left\|\phi_{s,a}^{s'} - \sum_{s'' \in \mathcal{S}_{h,s,a}} p_{s,a}^{s''}(\theta_h^*)\phi_{s,a}^{s''}\right\|_{\mathcal{H}_{k,h}^{-1}(\theta_h^*)},$$

$$\epsilon_{s,a}^{\mathtt{2nd}} = 90H\widehat{\beta}_k^2 \max_{s'}\|\phi_{s,a}^{s'}\|_{\mathcal{H}_{k,h}^{-1}(\theta_h^*)}^2.$$

By Lemma 9, we have

$$\sum_{k=1}^{K} V_1^*(s_{k,1}) - \sum_{k=1}^{K} V_1^{\pi_k}(s_{k,1}) \leq 2\sum_{k=1}^{K}\sum_{h=1}^{H}(\epsilon_{k,h}^{\mathtt{1st}} + \epsilon_{k,h}^{\mathtt{2nd}}) + H\sqrt{2KH\log(2/\delta)} \qquad (20)$$

Next, we bound $\epsilon_{k,h}^{\mathtt{1st}}$ and $\epsilon_{k,h}^{\mathtt{2nd}}$ respectively.

**Bounding $\epsilon_{k,h}^{\mathtt{1st}}$.** By statement (I) of Lemma 6, we have

$$\sum_{k=1}^{K}\sum_{s' \in \mathcal{S}_{k,h}} p_{k,h}^{s'}(\theta_h^*)\left\|\phi_{k,h}^{s'} - \sum_{s'' \in \mathcal{S}_{k,h}} p_{k,h}^{s''}(\theta_h^*)\phi_{k,h}^{s''}\right\|_{\mathcal{H}_{k,h}^{-1}(\theta_h^*)}$$

$$\leq \sqrt{\sum_{k=1}^{K}\sum_{s' \in \mathcal{S}_{k,h}} p_{k,h}^{s'}(\theta_h^*)}\sqrt{\sum_{k=1}^{K}\sum_{s' \in \mathcal{S}_{k,h}} p_{k,h}^{s'}(\theta_h^*)\left\|\phi_{k,h}^{s'} - \sum_{s'' \in \mathcal{S}_{k,h}} p_{k,h}^{s''}(\theta_h^*)\phi_{k,h}^{s''}\right\|_{\mathcal{H}_{k,h}^{-1}(\theta_h^*)}^2}$$

$$\leq \sqrt{2dK\log\left(1 + \frac{K}{d\lambda_K}\right)}$$

By the configuration that $\widehat{\beta}_k = (B + 3)\sqrt{d\log(k/\delta)}$, we have

$$\sum_{k=1}^{K}\sum_{h=1}^{H} \epsilon_{k,h}^{\mathtt{1st}} \leq (1 + 3\sqrt{2})(B + 3)dH^2\sqrt{K\log\left(1 + \frac{K}{d\lambda_K}\right)\log\left(\frac{k}{\delta}\right)}. \qquad (21)$$

**Bounding $\epsilon_{k,h}^{\mathtt{2nd}}$.** By statement (IV) of Lemma 6, we have

$$\sum_{k=1}^{K}\sum_{h=1}^{H} \epsilon_{k,h}^{\mathtt{2nd}} = 90H^2\sum_{k=1}^{K} \widehat{\beta}_k^2 \max_{s'}\|\phi_{s,a}^{s'}\|_{\mathcal{H}_{k,h}^{-1}(\theta_h^*)}^2$$

$$\leq \frac{180}{\kappa}H^2\widehat{\beta}_K^2 d\log\left(1 + \frac{K}{d\lambda_K}\right)$$

$$\leq \frac{180}{\kappa}(B + 3)^2 d^2 H^2 \log\left(1 + \frac{K}{d\lambda_K}\right)\log\left(\frac{K}{\delta}\right) \qquad (22)$$

where the first inequality holds by $sum_{i=1}^{k} \max_{s' \in \mathcal{S}_{i,h}}\|\phi_{i,h}^{s'}\|_{\mathcal{H}_{i,h}^{-1}(\theta)}^2 \leq \frac{2}{\kappa}d\log\left(1 + \frac{k}{\lambda_k d}\right), \forall\theta \in \Theta$ in Lemma 6 and the second inequality holds by the configuration of $\widehat{\beta}_k$.

Combining (20), (21), and (22), we have with probability at least $1 - \delta$,

$$\text{Reg}(K) \leq \sqrt{2dK\log\left(1 + \frac{K}{d\lambda_K}\right)} + \frac{180}{\kappa}(B + 3)^2 d^2 H^2 \log\left(1 + \frac{K}{d\lambda_K}\right)\log\left(\frac{K}{\delta}\right)$$

$$\leq \widetilde{\mathcal{O}}\left(dH^2\sqrt{K} + \kappa^{-1}d^2H^2\right).$$

This finishes the proof. ∎

# F  Omitted Proofs for Section 5

## F.1  Useful Lemma

**Lemma 11.** *For any $k \in [K]$, $h \in [H]$, define the second-order approximation of the loss function $\ell_{k,h}(\theta)$ at the estimator $\widetilde{\theta}_{k,h}$ as $\widetilde{\ell}_{k,h}(\theta) = \ell_{k,h}(\theta_{k,h}) + \langle \nabla\ell_{k,h}(\widetilde{\theta}_{k,h}), \theta - \widetilde{\theta}_{k,h}\rangle + \frac{1}{2}\|\theta - \widetilde{\theta}_{k,h}\|^2_{H_{k,h}(\widetilde{\theta}_{k,h})}$. Then, for the following update rule*

$$\widetilde{\theta}_{k+1,h} = \arg\min_{\theta \in \Theta} \widetilde{\ell}_{k,h}(\theta) + \frac{1}{2\eta}\|\theta - \widetilde{\theta}_{k,h}\|^2_{\mathcal{H}_{k,h}},$$

*it holds that*

$$\|\widetilde{\theta}_{k+1,h} - \theta^*_h\|^2_{\mathcal{H}_{k+1,h}} \leq 2\eta\left(\sum_{i=1}^{k}\ell_{i,h}(\theta^*_h) - \sum_{i=1}^{k}\ell_{i,h}(\widetilde{\theta}_{i+1,h})\right) + 4\lambda B$$

$$+ 12\sqrt{2}B\eta\sum_{i=1}^{k}\|\widetilde{\theta}_{i+1,h} - \widetilde{\theta}_{i,h}\|^2_2 - \sum_{i=1}^{k}\|\widetilde{\theta}_{i+1,h} - \widetilde{\theta}_{i,h}\|^2_{\mathcal{H}_{i,h}}.$$

*Proof.* Based on the analysis of (implicit) OMD update (see Lemma 16), for any $i \in [K]$, we have

$$\langle \nabla\widetilde{\ell}_{i,h}(\widetilde{\theta}_{i+1,h}), \widetilde{\theta}_{i+1,h} - \theta^*_h\rangle \leq \frac{1}{2\eta}\left(\|\widetilde{\theta}_{i,h} - \theta^*_h\|^2_{\mathcal{H}_{i,h}} - \|\widetilde{\theta}_{i+1,h} - \theta^*_h\|^2_{\mathcal{H}_{i,h}} - \|\widetilde{\theta}_{i+1,h} - \widetilde{\theta}_{i,h}\|^2_{\mathcal{H}_{i,h}}\right)$$

According to Lemma 4, we have

$$\ell_{i,h}(\widetilde{\theta}_{i+1,h}) - \ell_{i,h}(\theta^*_h) \leq \langle \nabla\ell_{i,h}(\widetilde{\theta}_{i+1,h}), \widetilde{\theta}_{i+1,h} - \theta^*_h\rangle - \frac{1}{\zeta}\left\|\widetilde{\theta}_{i+1,h} - \theta^*_h\right\|^2_{\nabla^2\ell_{i,h}(\widetilde{\theta}_{i+1,h})},$$

where $\zeta = \log(K+1) + 4$. Then, by combining the above two inequalities, we have

$$\ell_{i,h}(\widetilde{\theta}_{i+1,h}) - \ell_{i,h}(\theta^*_h) \leq \langle \nabla\ell_{i,h}(\widetilde{\theta}_{i+1,h}) - \nabla\widetilde{\ell}_{i,h}(\widetilde{\theta}_{i+1,h}), \widetilde{\theta}_{i+1,h} - \theta^*_h\rangle$$

$$+ \frac{1}{\zeta}\left(\|\widetilde{\theta}_{i,h} - \theta^*_h\|^2_{\mathcal{H}_{i,h}} - \|\widetilde{\theta}_{i+1,h} - \theta^*_h\|^2_{\mathcal{H}_{i+1,h}} - \|\widetilde{\theta}_{i+1,h} - \widetilde{\theta}_{i,h}\|^2_{\mathcal{H}_{i,h}}\right).$$

We can further bound the first term of the right-hand side as:

$$\langle \nabla\ell_{i,h}(\widetilde{\theta}_{i+1,h}) - \nabla\widetilde{\ell}_{i,h}(\widetilde{\theta}_{i+1,h}), \widetilde{\theta}_{i+1,h} - \theta^*_h\rangle$$

$$= \langle \nabla\ell_{i,h}(\widetilde{\theta}_{i+1,h}) - \nabla\ell_{i,h}(\widetilde{\theta}_{i,h}) - \nabla^2\ell_{i,h}(\widetilde{\theta}_{i,h})(\widetilde{\theta}_{i+1,h} - \widetilde{\theta}_{i,h}), \widetilde{\theta}_{i+1,h} - \theta^*_h\rangle$$

$$= \langle D^3\ell_{i,h}(\xi_{i+1})[\widetilde{\theta}_{i+1,h} - \widetilde{\theta}_{i,h}](\widetilde{\theta}_{i+1,h} - \widetilde{\theta}_{i,h}), \widetilde{\theta}_{i+1,h} - \theta^*_h\rangle$$

$$\leq 3\sqrt{2}\|\widetilde{\theta}_{i+1,h} - \theta^*_h\|_2\|\widetilde{\theta}_{i+1,h} - \widetilde{\theta}_{i,h}\|^2_{\nabla^2\ell_{i,h}(\xi_{i+1})}$$

$$\leq 6\sqrt{2}B\|\widetilde{\theta}_{i+1,h} - \widetilde{\theta}_{i,h}\|^2_2,$$

where the second equality holds by the mean value theorem, the first inequality holds by the self-concordant-like property of $\ell_{i,h}(\cdot)$ in Proposition 1, and the last inequality holds by $\widetilde{\theta}_{i+1,h}$ and $\theta^*_h$ belong to $\Theta = \{\theta \in \mathbb{R}^d, \|\theta\|_2 \leq B\}$, and $\nabla^2\ell_{i,h}(\xi_{i+1}) \preceq I_d$.

Then, by taking the summation over $i$ and rearranging the terms, we obtain

$$\|\widetilde{\theta}_{k+1,h} - \theta^*_h\|^2_{\mathcal{H}_{k+1,h}} \leq \zeta\left(\sum_{s=1}^{k}\ell_{s,h}(\theta^*_h) - \sum_{s=1}^{k}\ell_{s,h}(\widetilde{\theta}_{i+1,h})\right) + \|\widetilde{\theta}_{1,h} - \theta^*_h\|^2_{\mathcal{H}_{1,h}}$$

$$+ 6\sqrt{2}B\zeta\sum_{s=1}^{k}\|\widetilde{\theta}_{i+1,h} - \widetilde{\theta}_{i,h}\|^2_2 - \sum_{s=1}^{k}\|\widetilde{\theta}_{i+1,h} - \widetilde{\theta}_{i,h}\|^2_{\mathcal{H}_{i,h}}$$

$$\leq \zeta\left(\sum_{s=1}^{k}\ell_{s,h}(\theta^*_h) - \sum_{s=1}^{k}\ell_{s,h}\left(\widetilde{\theta}_{i+1,h}\right)\right) + 4\lambda B$$

$$+ 6\sqrt{2}B\zeta\sum_{s=1}^{k}\|\widetilde{\theta}_{i+1,h} - \widetilde{\theta}_{i,h}\|^2_2 - \sum_{s=1}^{k}\|\widetilde{\theta}_{i+1,h} - \widetilde{\theta}_{i,h}\|^2_{\mathcal{H}_{i,h}},$$

where the last inequality holds by $\|\widetilde{\theta}_{1,h} - \theta^*_h\|^2_{\mathcal{H}_{1,h}} \leq \lambda\|\widetilde{\theta}_{1,h} - \theta^*_h\|^2_2 \leq 4\lambda B$. Plugging $\zeta = 2\eta$ finishes the proof. ∎

## F.2 Proof of Lemma 2

*Proof.* According to Lemma 11, we have

$$\|\widetilde{\theta}_{k+1,h} - \theta_h^*\|_{\mathcal{H}_{k+1,h}}^2 \leq 2\eta \left( \sum_{i=1}^{k} \ell_{i,h}(\theta_h^*) - \sum_{i=1}^{k} \ell_{i,h}(\widetilde{\theta}_{i+1,h}) \right) + 4\lambda B$$
$$+ 12\sqrt{2}B\eta \sum_{i=1}^{k} \|\widetilde{\theta}_{i+1,h} - \widetilde{\theta}_{i,h}\|_2^2 - \sum_{i=1}^{k} \|\widetilde{\theta}_{i+1,h} - \widetilde{\theta}_{i,h}\|_{\mathcal{H}_{i,h}}^2.$$

We bound the right-hand side of the above lemma separately in the following. The most challenging part is to bound the term $\sum_{i=1}^{k} \ell_{i,h}(\theta_h^*) - \sum_{i=1}^{k} \ell_{i,h}(\widetilde{\theta}_{i+1,h})$. At first glance, this term appears straightforward to control, as it can be observed that $\theta_h^* = \arg\min_{\theta \in \mathbb{R}^d} \bar{\ell}_h(\theta) \triangleq \mathbb{E}y_{i,h}[\ell_{i,h}(\theta)]$, where the instantaneous loss $\ell_{i,h}(\theta)$ serves as an empirical observation of $\bar{\ell}_h(\theta)$. Consequently, the loss gap term seemingly can be bounded using appropriate concentration results. However, a caveat lies in the fact that the update of the estimator $\widetilde{\theta}_{i+1,h}$ depends on the information $\ell_{i,h}$, or more precisely $y_{i,h}$, making it difficult to directly apply such concentration results.

To address this issue, we decompose the loss gap into two components by introducing an intermediate term. Specifically, we define the softmax function as $[\sigma_{k,h}(z)]_s = \frac{\exp([z])_s}{1 + \sum_{s \in \dot{\mathcal{S}}_{k,h}} \exp([z])_s}, \forall s \in \mathcal{S}_{k,h}$. Using this definition, the loss function can be rewritten as:

$$\ell_{k,h}(z_{k,h}, y_{k,h}) = \sum_{s' \in \mathcal{S}_{k,h}} \mathbb{1}[y_{k,h}^{s'} = 1] \log \left( \frac{1}{[\sigma_{k,h}(z_{k,h})]_{s'}} \right).$$

Define a pseudo-inverse function of $\sigma_{k,h}(\cdot)$ as $[\sigma_{k,h}^{-1}(p)]_{s'} = \log \left( \frac{[p]_{s'}}{1 - \|p\|_1} \right), \forall p \in \{p \in [0,1]^{S_{k,h}} \mid \|p\|_1 < 1\}$. Then, the loss gap term can be decomposed into two parts as follows.

$$\sum_{i=1}^{k} \left( \ell_{i,h}(\theta_h^*) - \ell_{i,h}(\widetilde{\theta}_{i+1,h}) \right)$$
$$= \underbrace{\sum_{i=1}^{k} (\ell_{i,h}(\theta_h^*) - \ell_{i,h}(z_{i,h}, y_{i,h}))}_{\texttt{term (a)}} + \underbrace{\sum_{i=1}^{k} \left( \ell_{i,h}(z_{i,h}, y_{i,h}) - \ell_{i,h}(\widetilde{\theta}_{i+1,h}) \right)}_{\texttt{term (b)}}$$

where $z_{k,h} = \sigma_{k,h}^{-1}(\mathbb{E}_{\theta \sim P_{k,h}}[\sigma_{k,h}((\phi_{k,h}^{s'})^\top \theta)_{s' \in \dot{\mathcal{S}}_{k,h}}])$, $P_{k,h} \triangleq \mathcal{N}(\widetilde{\theta}_{k,h}, (1 + c\mathcal{H}_{k,h}^{-1}))$ is the Gaussian distribution with mean $\widetilde{\theta}_{k,h}$ and covariance $(1 + c\mathcal{H}_{k,h}^{-1})$ where $c$ is a constant to be specified later.

The design of the intermediate term was originally proposed by Zhang and Sugiyama [2023] in their study of the multiclass logistic bandit problem and was subsequently applied to the multinomial logit bandits problem by Lee and Oh [2024]. Notably, the intermediate loss is independent of the information contained in $y_{i,h}$, enabling the application of concentration results. Specifically, based on Lemma F.2 and Lemma F.3 of Lee and Oh [2024], we obtain the following upper bounds for them.

For term (a), let $\delta \in (0,1]$ and $\lambda \geq \max\{2, 72cd\}$, for all $k \in [K], h \in [H]$, with probability at least $1 - \delta$, we have

$\texttt{term (a)}$

$$\leq (3\log(1 + (U+1)k) + 3) \left( \frac{17}{16}\lambda + 2\sqrt{\lambda}\log\left( \frac{2H\sqrt{1+2k}}{\delta} \right) + 16\left( \log\left( \frac{2H\sqrt{1+2k}}{\delta} \right) \right)^2 \right) + 2.$$

For term (b), for all $k \in [K], h \in [H]$, we have

$$\texttt{term (b)} \leq \frac{1}{2c} \sum_{i=1}^{k} \left\| \widetilde{\theta}_{i,h} - \theta_{i+1,h} \right\|_{\mathcal{H}_{i,h}}^2 + \sqrt{6}cd \log\left( 1 + \frac{k+1}{2\lambda} \right).$$

Combining term (a) and term (b), we have

$$\|\widetilde{\theta}_{k+1,h} - \theta_h^*\|_{\mathcal{H}_{k+1,h}}^2$$

$$\leq 2\eta\left[(3\log(1+(U+1)k)+3)\left(\frac{17}{16}\lambda + 2\sqrt{\lambda}\log\left(\frac{2H\sqrt{1+2k}}{\delta}\right) + 16\left(\log\left(\frac{2H\sqrt{1+2k}}{\delta}\right)\right)^2\right)\right.$$

$$+2+\sqrt{6}cd\log\left(1+\frac{k+1}{2\lambda}\right)\Bigg] + 4\lambda B + 12\sqrt{2}B\eta\sum_{i=1}^{k}\|\widetilde{\theta}_{i+1,h}-\widetilde{\theta}_{i,h}\|_2^2 + \left(\frac{\eta}{c}-1\right)\sum_{i=1}^{k}\|\widetilde{\theta}_{i+1,h}+\widetilde{\theta}_{i,h}\|_{\mathcal{H}_{i,h}}^2$$

$$\leq 2\eta\left[(3\log(1+(U+1)k)+3)\left(\frac{17}{16}\lambda + 2\sqrt{\lambda}\log\left(\frac{2H\sqrt{1+2k}}{\delta}\right) + 16\left(\log\left(\frac{2H\sqrt{1+2k}}{\delta}\right)\right)^2\right)\right.$$

$$+ 2 + \sqrt{6}cd\log\left(1+\frac{k+1}{2\lambda}\right)\Bigg] + 4\lambda B,$$

where the second inequality holds by setting $c = 7\eta/6$ and $\lambda \geq \max\{84\sqrt{2}\eta B, 84d\eta\}$, we have

$$12\sqrt{2}B\eta\sum_{i=1}^{k}\|\widetilde{\theta}_{i+1,h}-\widetilde{\theta}_{i,h}\|_2^2 + \left(\frac{\eta}{c}-1\right)\sum_{i=1}^{k}\|\widetilde{\theta}_{i+1,h}+\widetilde{\theta}_{i,h}\|_{\mathcal{H}_{i,h}}^2$$

$$= 12\sqrt{2}B\eta\sum_{i=1}^{k}\|\widetilde{\theta}_{i+1,h}-\widetilde{\theta}_{i,h}\|_2^2 - \frac{1}{7}\sum_{i=1}^{k}\|\widetilde{\theta}_{i+1,h}+\widetilde{\theta}_{i,h}\|_{\mathcal{H}_{i,h}}^2$$

$$\leq \left(12\sqrt{2}B\eta - \frac{\lambda}{7}\right)\sum_{i=1}^{k}\|\widetilde{\theta}_{i+1,h}-\widetilde{\theta}_{i,h}\|_2^2$$

$$\leq 0.$$

Thus, by setting $\eta = \frac{1}{2}\log(U+1) + (B+1)$, $\lambda = 84\sqrt{2}\eta(B+d)$, we have

$$\|\widetilde{\theta}_{k+1,h} - \theta_h^*\|_{\mathcal{H}_{k+1,h}} \leq \mathcal{O}\big(\sqrt{d}\log U \log(kH/\delta)\big) \triangleq \widetilde{\beta}_k.$$

This finishes the proof. ∎

### F.3 Proof of Lemma 3

*Proof.* Lemma 3 follows directly by substituting the confidence set $\widehat{\mathcal{C}}_{k,h}$ defined in Lemma 2, into Lemma 7. This finishes the proof. ∎

### F.4 Proof of Theorem 2

*Proof.* Combining Lemma 3 and Lemma 9, we have

$$\sum_{k=1}^{K}V_1^*(s_{k,1}) - \sum_{k=1}^{K}V_1^{\pi_k}(s_{k,1}) \leq 2\sum_{k=1}^{K}\sum_{h=1}^{H}(\epsilon_{k,h}^{\mathtt{fst}} + \epsilon_{k,h}^{\mathtt{snd}}) + H\sqrt{2KH\log(2/\delta)}$$

where

$$\epsilon_{k,h}^{\mathtt{fst}} = H\widetilde{\beta}_k\sum_{s'\in\mathcal{S}_{k,h}}p_{k,h}^{s'}(\widetilde{\theta}_{k,h})\left\|\phi_{k,h}^{s'} - \sum_{s''\in\mathcal{S}_{k,h}}p_{k,h}^{s''}(\widetilde{\theta}_{k,h})\phi_{k,h}^{s''}\right\|_{\mathcal{H}_{k,h}^{-1}},$$

$$\epsilon_{k,h}^{\mathtt{snd}} = \frac{5}{2}H\widetilde{\beta}_k^2\max_{s'\in\mathcal{S}_{k,h}}\|\phi_{k,h}^{s'}\|_{\mathcal{H}_{k,h}^{-1}}^2.$$

Next, we bound $\epsilon_{k,h}^{\mathtt{fst}}$ and $\epsilon_{k,h}^{\mathtt{snd}}$ respectively.

**Bounding $\epsilon_{k,h}^{\mathtt{fst}}$.** For simplicity, we denote

$$\mathbb{E}_\theta[\phi_{k,h}^{s'}] = \mathbb{E}_{s'\sim p_{k,h}^s(\theta)}[\phi_{k,h}^{s'}], \quad \bar{\phi}_{s,a}^{s'} = \phi_{s,a}^{s'} - \mathbb{E}_{\widetilde{\theta}_{k,h}}[\phi_{k,h}^{s'}], \quad \widetilde{\phi}_{s,a}^{s'} = \phi_{s,a}^{s'} - \mathbb{E}_{\widetilde{\theta}_{k+1,h}}[\phi_{k,h}^{s'}]$$

Then, we have

$$
\Big\| \sum_{s' \in \mathcal{S}_{k,h}} p_{k,h}^{s'}(\widetilde{\theta}_{k,h}) \big\| \phi_{k,h}^{s'} - \sum_{s'' \in \mathcal{S}_{k,h}} p_{k,h}^{s''}(\widetilde{\theta}_{k,h}) \phi_{k,h}^{s''} \big\|_{\mathcal{H}_{k,h}^{-1}} = \sum_{s' \in \mathcal{S}_{k,h}} p_{k,h}^{s'}(\widetilde{\theta}_{k,h}) \big\| \bar{\phi}_{k,h}^{s'} \big\|_{\mathcal{H}_{k,h}^{-1}}
$$

$$
\leq \sum_{s' \in \mathcal{S}_{k,h}} p_{k,h}^{s'}(\widetilde{\theta}_{k,h}) \big\| \bar{\phi}_{k,h}^{s'} - \widetilde{\phi}_{k,h}^{s'} \big\|_{\mathcal{H}_{k,h}^{-1}} + \sum_{s' \in \mathcal{S}_{k,h}} p_{k,h}^{s'}(\widetilde{\theta}_{k,h}) \big\| \widetilde{\phi}_{k,h}^{s'} \big\|_{\mathcal{H}_{k,h}^{-1}}
$$

$$
= \underbrace{\sum_{s' \in \mathcal{S}_{k,h}} p_{k,h}^{s'}(\widetilde{\theta}_{k,h}) \big\| \bar{\phi}_{k,h}^{s'} - \widetilde{\phi}_{k,h}^{s'} \big\|_{\mathcal{H}_{k,h}^{-1}}}_{\texttt{term (c)}} + \underbrace{\sum_{s' \in \mathcal{S}_{k,h}} (p_{k,h}^{s'}(\widetilde{\theta}_{k,h}) - p_{k,h}^{s'}(\widetilde{\theta}_{k+1,h})) \big\| \widetilde{\phi}_{k,h}^{s'} \big\|_{\mathcal{H}_{k,h}^{-1}}}_{\texttt{term (d)}}
$$

$$
+ \underbrace{\sum_{s' \in \mathcal{S}_{k,h}} p_{k,h}^{s'}(\widetilde{\theta}_{k+1,h}) \big\| \widetilde{\phi}_{k,h}^{s'} \big\|_{\mathcal{H}_{k,h}^{-1}}}_{\texttt{term (e)}} .
$$

We bound these terms separately in the following.

For the first term $(c)$, we have

$$
\big\| \bar{\phi}_{k,h}^{s'} - \widetilde{\phi}_{k,h}^{s'} \big\|_{\mathcal{H}_{k,h}^{-1}}
$$

$$
= \Big\| \sum_{s'' \in \mathcal{S}_{k,h}} \left( p_{k,h}^{s''}(\widetilde{\theta}_{k+1,h}) - p_{k,h}^{s''}(\widetilde{\theta}_{k,h}) \right) \phi_{k,h}^{s''} \Big\|_{\mathcal{H}_{k,h}^{-1}}
$$

$$
= \Big\| \sum_{s'' \in \mathcal{S}_{k,h}} \left( \nabla p_{k,h}^{s''}(\xi_{k,h})^\top (\widetilde{\theta}_{k+1,h} - \widetilde{\theta}_{k,h}) \right) \phi_{k,h}^{s''} \Big\|_{\mathcal{H}_{k,h}^{-1}}
$$

$$
\leq \sum_{s'' \in \mathcal{S}_{k,h}} \left| \nabla p_{k,h}^{s''}(\xi_{k,h})^\top (\widetilde{\theta}_{k+1,h} - \widetilde{\theta}_{k,h}) \right| \cdot \big\| \phi_{k,h}^{s''} \big\|_{\mathcal{H}_{k,h}^{-1}}
$$

$$
= \sum_{s'' \in \mathcal{S}_{k,h}} \Big| \Big( p_{k,h}^{s''}(\xi_{k,h}) \phi_{k,h}^{s''} - p_{k,h}^{s''}(\xi_{k,h}) \sum_{s''' \in \mathcal{S}_{k,h}} p_{k,h}^{s'''}(\xi_{k,h}) \phi_{k,h}^{s'''} \Big)^\top (\widetilde{\theta}_{k+1,h} - \widetilde{\theta}_{k,h}) \Big| \cdot \big\| \phi_{k,h}^{s''} \big\|_{\mathcal{H}_{k,h}^{-1}}
$$

$$
\leq \sum_{s'' \in \mathcal{S}_{k,h}} p_{k,h}^{s''}(\xi_{k,h}) \Big| (\phi_{k,h}^{s''})^\top (\widetilde{\theta}_{k+1,h} - \widetilde{\theta}_{k,h}) \Big| \cdot \big\| \phi_{k,h}^{s''} \big\|_{\mathcal{H}_{k,h}^{-1}}
$$

$$
+ \sum_{s'' \in \mathcal{S}_{k,h}} p_{k,h}^{s''}(\xi_{k,h}) \big\| \phi_{k,h}^{s''} \big\|_{\mathcal{H}_{k,h}^{-1}} \sum_{s''' \in \mathcal{S}_{k,h}} p_{k,h}^{s'''}(\xi_{k,h}) \cdot \Big| (\phi_{k,h}^{s'''})^\top (\widetilde{\theta}_{k+1,h} - \widetilde{\theta}_{k,h}) \Big|
$$

$$
\leq \sum_{s'' \in \mathcal{S}_{k,h}} p_{k,h}^{s''}(\xi_{k,h}) \big\| \widetilde{\theta}_{k+1,h} - \widetilde{\theta}_{k,h} \big\|_{\mathcal{H}_{k,h}} \big\| \phi_{k,h}^{s''} \big\|_{\mathcal{H}_{k,h}^{-1}}^2
$$

$$
+ \sum_{s'' \in \mathcal{S}_{k,h}} p_{k,h}^{s''}(\xi_{k,h}) \big\| \phi_{k,h}^{s''} \big\|_{\mathcal{H}_{k,h}^{-1}} \sum_{s''' \in \mathcal{S}_{k,h}} p_{k,h}^{s'''}(\xi_{k,h}) \big\| \phi_{k,h}^{s'''} \big\|_{\mathcal{H}_{k,h}^{-1}} \big\| \widetilde{\theta}_{k+1,h} - \widetilde{\theta}_{k,h} \big\|_{\mathcal{H}_{k,h}}
$$

$$
\leq \frac{4\eta}{\sqrt{\lambda}} \sum_{s'' \in \mathcal{S}_{k,h}} p_{k,h}^{s''}(\xi_{k,h}) \big\| \phi_{k,h}^{s''} \big\|_{\mathcal{H}_{k,h}^{-1}}^2 + \frac{4\eta}{\sqrt{\lambda}} \Big( \sum_{s'' \in \mathcal{S}_{k,h}} p_{k,h}^{s''}(\xi_{k,h}) \big\| \phi_{k,h}^{s''} \big\|_{\mathcal{H}_{k,h}^{-1}} \Big)^2
$$

$$
\leq \frac{8\eta}{\sqrt{\lambda}} \sum_{s'' \in \mathcal{S}_{k,h}} p_{k,h}^{s''}(\xi_{k,h}) \big\| \phi_{k,h}^{s''} \big\|_{\mathcal{H}_{k,h}^{-1}}^2
$$

$$
\leq \frac{8\eta}{\sqrt{\lambda}} \max_{s'' \in \mathcal{S}_{k,h}} \big\| \phi_{k,h}^{s''} \big\|_{\mathcal{H}_{k,h}^{-1}}^2
$$

where and the fourth inequality is because by Lemma 15 and the fact $\widetilde{\mathcal{H}}_{k,h} \succeq \mathcal{H}_{k,h} \succeq \lambda I_d$, we have

$$
\big\| \widetilde{\theta}_{k+1,h} - \widetilde{\theta}_{k,h} \big\|_{\mathcal{H}_{k,h}} \leq \big\| \widetilde{\theta}_{k+1,h} - \widetilde{\theta}_{k,h} \big\|_{\widetilde{\mathcal{H}}_{k,h}} \leq 2\eta \| \nabla \ell_{k,h}(\widetilde{\theta}_{k,h}) \|_{\widetilde{\mathcal{H}}_{k,h}^{-1}} \leq \frac{2\eta}{\sqrt{\lambda}} \| \nabla \ell_{k,h}(\widetilde{\theta}_{k,h}) \|_2,
$$

and since $\nabla \ell_{k,h}(\theta) = \sum_{s' \in \mathcal{S}_{k,h}} (p_{k,h}^{s'}(\theta) - y_{k,h}^{s'}) \phi_{k,h}^{s'}$, we have

$$
\| \nabla \ell_{k,h}(\widetilde{\theta}_{k,h}) \|_2 \leq \Big\| \sum_{s' \in \mathcal{S}_{k,h}} p_{k,h}^{s'}(\widetilde{\theta}_{k,h}) \phi_{k,h}^{s'} \Big\|_2 + \Big\| \sum_{s' \in \mathcal{S}_{k,h}} y_{k,h}^{s'} \phi_{k,h}^{s'} \Big\|_2 \leq 2 \max_{s' \in \mathcal{S}_{k,h}} \big\| \phi_{k,h}^{s'} \big\|_2 \leq 2.
$$

Therefore, we have

$$\sum_{k=1}^{K}\sum_{h=1}^{H}\sum_{s'\in\mathcal{S}_{k,h}} p_{k,h}^{s'}(\widetilde{\theta}_{k,h})\|\bar{\phi}_{k,h}^{s'}-\widetilde{\phi}_{k,h}^{s'}\|_{\mathcal{H}_{k,h}^{-1}} \leq \frac{8\eta}{\sqrt{\lambda}}\sum_{k=1}^{K}\sum_{h=1}^{H}\sum_{s'\in\mathcal{S}_{k,h}} p_{k,h}^{s'}(\widetilde{\theta}_{k,h})\max_{s''\in\mathcal{S}_{k,h}}\|\phi_{k,h}^{s''}\|_{\mathcal{H}_{k,h}^{-1}}^{2}$$

$$\leq \frac{8\eta}{\sqrt{\lambda}}\sum_{k=1}^{K}\sum_{h=1}^{H}\sum_{s'\in\mathcal{S}_{k,h}}\max_{s''\in\mathcal{S}_{k,h}}\|\phi_{k,h}^{s''}\|_{\mathcal{H}_{k,h}^{-1}}^{2}$$

$$\leq \frac{16H\eta}{\kappa\sqrt{\lambda}}d\log\left(1+\frac{K}{d\lambda}\right), \tag{23}$$

where the last inequality holds by $\sum_{i=1}^{k}\max_{s'\in\mathcal{S}_{i,h}}\|\phi_{i,h}^{s'}\|_{\mathcal{H}_{i,h}^{-1}}^{2} \leq \frac{2}{\kappa}d\log\left(1+\frac{k}{\lambda_k d}\right)$ in Lemma 6.

For the term $(d)$, by similar analysis, we have

$$(p_{k,h}^{s'}(\widetilde{\theta}_{k,h})-p_{k,h}^{s'}(\widetilde{\theta}_{k+1,h}))\|\widetilde{\phi}_{k,h}^{s'}\|_{\mathcal{H}_{k,h}^{-1}}$$

$$= \nabla p_{k,h}^{s'}(\xi_{k,h})^{\top}(\widetilde{\theta}_{k,h}-\widetilde{\theta}_{k+1,h})\|\widetilde{\phi}_{k,h}^{s'}\|_{\mathcal{H}_{k,h}^{-1}}$$

$$= \left(p_{k,h}^{s'}(\xi_{k,h})\phi_{k,h}^{s'}-p_{k,h}^{s'}(\xi_{k,h})\sum_{s''\in\mathcal{S}_{k,h}}p_{k,h}^{s''}(\xi_{k,h})\phi_{k,h}^{s''}\right)^{\top}(\widetilde{\theta}_{k+1,h}-\widetilde{\theta}_{k,h})\|\widetilde{\phi}_{k,h}^{s'}\|_{\mathcal{H}_{k,h}^{-1}}$$

$$\leq \frac{4\eta}{\sqrt{\lambda}}\left(p_{k,h}^{s'}(\xi_{k,h})\|\phi_{k,h}^{s'}\|_{\mathcal{H}_{k,h}^{-1}}\|\widetilde{\phi}_{k,h}^{s'}\|_{\mathcal{H}_{k,h}^{-1}}+p_{k,h}^{s'}(\xi_{k,h})\|\widetilde{\phi}_{k,h}^{s'}\|_{\mathcal{H}_{k,h}^{-1}}\sum_{s''\in\mathcal{S}_{k,h}}p_{k,h}^{s''}(\xi_{k,h})\|\phi_{k,h}^{s''}\|_{\mathcal{H}_{k,h}^{-1}}\right)$$

$$\leq \frac{4\eta}{\sqrt{\lambda}}\left(\max_{s''\in\mathcal{S}_{k,h}}\|\phi_{k,h}^{s''}\|_{\mathcal{H}_{k,h}^{-1}}\|\widetilde{\phi}_{k,h}^{s''}\|_{\mathcal{H}_{k,h}^{-1}}+\max_{s''\in\mathcal{S}_{k,h}}\|\widetilde{\phi}_{k,h}^{s''}\|_{\mathcal{H}_{k,h}^{-1}}\max_{s'''\in\mathcal{S}_{k,h}}\|\phi_{k,h}^{s'''}\|_{\mathcal{H}_{k,h}^{-1}}\right)$$

$$\leq \frac{2\eta}{\sqrt{\lambda}}\max_{s''\in\mathcal{S}_{k,h}}\left(\|\phi_{k,h}^{s''}\|_{\mathcal{H}_{k,h}^{-1}}^{2}+\|\widetilde{\phi}_{k,h}^{s''}\|_{\mathcal{H}_{k,h}^{-1}}^{2}\right)$$

$$\qquad + \frac{2\eta}{\sqrt{\lambda}}\left(\left(\max_{s''\in\mathcal{S}_{k,h}}\|\widetilde{\phi}_{k,h}^{s''}\|_{\mathcal{H}_{k,h}^{-1}}\right)^{2}+\left(\max_{s'''\in\mathcal{S}_{k,h}}\|\phi_{k,h}^{s'''}\|_{\mathcal{H}_{k,h}^{-1}}\right)^{2}\right)$$

$$\leq \frac{8\eta}{\sqrt{\lambda}}\max\left\{\max_{s''\in\mathcal{S}_{k,h}}\|\phi_{k,h}^{s''}\|_{\mathcal{H}_{k,h}^{-1}}^{2},\max_{s''\in\mathcal{S}_{k,h}}\|\widetilde{\phi}_{k,h}^{s''}\|_{\mathcal{H}_{k,h}^{-1}}^{2}\right\},$$

where the third inequality holds by the AM-GM inequality. Thus, we have

$$\sum_{k=1}^{K}\sum_{h=1}^{H}\sum_{s'\in\mathcal{S}_{k,h}}(p_{k,h}^{s'}(\widetilde{\theta}_{k,h})-p_{k,h}^{s'}(\widetilde{\theta}_{k+1,h}))\|\widetilde{\phi}_{k,h}^{s'}\|_{\mathcal{H}_{k,h}^{-1}}$$

$$\leq \frac{8\eta}{\sqrt{\lambda}}\sum_{k=1}^{K}\sum_{h=1}^{H}\max\left\{\max_{s''\in\mathcal{S}_{k,h}}\|\phi_{k,h}^{s''}\|_{\mathcal{H}_{k,h}^{-1}}^{2},\max_{s''\in\mathcal{S}_{k,h}}\|\widetilde{\phi}_{k,h}^{s''}\|_{\mathcal{H}_{k,h}^{-1}}^{2}\right\}$$

$$\leq \frac{16H\eta}{\kappa\sqrt{\lambda}}d\log\left(1+\frac{K}{d\lambda}\right), \tag{24}$$

where the last inequality holds by $\sum_{i=1}^{k}\max_{s'\in\mathcal{S}_{i,h}}\|\widetilde{\phi}_{i,h}^{s'}\|_{\mathcal{H}_{i,h}^{-1}}^{2} \leq \frac{2}{\kappa}d\log\left(1+\frac{k}{\lambda_k d}\right)$ in Lemma 6.

Finally, we bound the term $(e)$ as follows.

$$\sum_{k=1}^{K}\sum_{s'\in\mathcal{S}_{k,h}}p_{k,h}^{s'}(\widetilde{\theta}_{k+1,h})\|\widetilde{\phi}_{k,h}^{s'}\|_{\mathcal{H}_{k,h}^{-1}}$$

$$\leq \sqrt{\sum_{k=1}^{K}\sum_{s'\in\mathcal{S}_{k,h}}p_{k,h}^{s'}(\widetilde{\theta}_{k+1,h})}\sqrt{\sum_{k=1}^{K}\sum_{s'\in\mathcal{S}_{k,h}}p_{k,h}^{s'}(\widetilde{\theta}_{k+1,h})\|\widetilde{\phi}_{k,h}^{s'}\|_{\mathcal{H}_{k,h}^{-1}}^{2}}$$

$$\leq \sqrt{K}\sqrt{2d\log\left(1+\frac{K}{d\lambda}\right)}, \tag{25}$$

where the first inequality holds by the Cauchy-Schwarz inequality and the last holds by Lemma 6. Thus, combining (23), (24), and (25), we have

$$\sum_{k=1}^{K}\sum_{h=1}^{H}\epsilon_{k,h}^{\texttt{fst}} = H\widetilde{\beta}_{K}\sum_{k=1}^{K}\sum_{h=1}^{H}\sum_{s'\in\mathcal{S}_{k,h}}p_{k,h}^{s'}(\widetilde{\theta}_{k,h})\Big\|\phi_{k,h}^{s'} - \sum_{s''\in\mathcal{S}_{k,h}}p_{k,h}^{s''}(\widetilde{\theta}_{k,h})\phi_{k,h}^{s''}\Big\|_{\mathcal{H}_{k,h}^{-1}}$$

$$\leq H^{2}\widetilde{\beta}_{K}\left(\sqrt{2dK\log\left(1+\frac{K}{d\lambda}\right)} + \frac{32\eta}{\kappa\sqrt{\lambda}}d\log\left(1+\frac{K}{d\lambda}\right)\right) \quad (26)$$

For the second-order term, by Lemma 6, we have

$$\sum_{k=1}^{K}\sum_{h=1}^{H}\epsilon_{k,h}^{\texttt{snd}} = \frac{5}{2}H\widetilde{\beta}_{k}^{2}\sum_{k=1}^{K}\sum_{h=1}^{H}\max_{s'\in\mathcal{S}_{k,h}}\|\phi_{k,h}^{s'}\|_{\mathcal{H}_{k,h}^{-1}}^{2} \leq \frac{5}{\kappa}H^{2}\widetilde{\beta}_{K}^{2}d\log\left(1+\frac{K}{d\lambda}\right). \quad (27)$$

where the last inequality holds by $\sum_{i=1}^{k}\max_{s'\in\mathcal{S}_{i,h}}\|\phi_{i,h}^{s'}\|_{\mathcal{H}_{i,h}^{-1}}^{2} \leq \frac{2}{\kappa}d\log\left(1+\frac{k}{\lambda_{k}d}\right)$ in Lemma 6.

Combining (26) and (27), we have

$$\text{Reg}(K) \leq 2\sum_{k=1}^{K}\sum_{h=1}^{H}(\epsilon_{k,h}^{\texttt{fst}} + \epsilon_{k,h}^{\texttt{snd}}) + H\sqrt{2KH\log(2/\delta)}$$

$$\leq H^{2}\widetilde{\beta}_{K}\left(\sqrt{2dK\log\left(1+\frac{K}{d\lambda}\right)} + \frac{32\eta}{\kappa\sqrt{\lambda}}d\log\left(1+\frac{K}{d\lambda}\right)\right) + \frac{5}{\kappa}H^{2}\widetilde{\beta}_{K}^{2}d\log\left(1+\frac{K}{d\lambda}\right)$$

$$+ H\sqrt{2KH\log\frac{2}{\delta}}$$

$$\leq \widetilde{\mathcal{O}}\left(dH^{2}\sqrt{K} + d^{2}H^{2}\kappa^{-1}\right).$$

This finishes the proof. ∎

## G  Omitted Proofs for Section 6

### G.1  Proof of Theorem 3

*Proof.* Our proof is similar to adversarial linear mixture MDPs with the unknown transition in Zhao et al. [2023]. We prove this lemma by reducing MNL mixture MDPs to a sequence of logistic bandits.

We use each three layers to construct a block. Note that the third layer of block $i$ is also the first layer of block $i+1$ and hence there are total $H/2$ blocks. In each block, both the first and third layers of this block only have one state, and the second layer has two states. Here we take block $i$ as an example. The first two layers of this block are associated with transition probability $\mathbb{P}_{i,1}$ and $\mathbb{P}_{i,2}$. Denote by $s_{i,1}$ the only state in the first layer of this block. In the second layer of the block $i$, we assume there exist two states $s_{i,2}^{*}$ and $s_{i,2}$. Let $s_{i,3}$ be the only state in the third layer of this block. Further, for any $a \in \mathcal{A}$, let $\phi(s_{i,1}, a, s_{i,2}) = 0$. The transition probability is defined as follows:

$$\mathbb{P}_{i,1}(s_{i,2}^{*} \mid s_{i,1}, a) = \frac{\exp(\phi(s_{i,2}^{*} \mid s_{i,1}, a)^{\top}\theta_{i,1}^{*})}{1 + \exp(\phi(s_{i,2}^{*} \mid s_{i,1}, a)^{\top}\theta_{i,1}^{*})} = \rho_{a}, \quad \mathbb{P}_{i,1}(s_{i,2} \mid s_{i,1}, a) = 1 - \rho_{a}.$$

For the second layer, it satisfies $\forall s = s_{i,2}^{*}, s_{i,2}$, and $a \in \mathcal{A}$, $\mathbb{P}_{i,2}(s_{i,3} \mid s, a) = 1$. The reward satisfies $r_{k}(s_{i,1}, a) = 0$ for the first layer and $r_{k}(s_{i,2}^{*}, a) = 1$, $r_{k}(s_{i,2}, a) = 0$ for the second for all $a \in \mathcal{A}$.

Then, consider the logistic bandit problem where a learner selects action $x \in \mathbb{R}^{d}$ and receives a reward $r_{k}$ sampled from the Bernoulli distribution with mean $\mu(x^{\top}\theta^{*}) = (1 + \exp(x^{\top}\theta^{*}))^{-1}$. By this configuration, we can see that learning in each block of MDP can be regarded as learning a $d$-dimensional logistic bandit problem with $A$ arms, where the arm set is $\phi(s_{i,2}^{*} \mid s_{i,1}, a)$ and the expected reward of each arm is $\rho_{a}$. Thus, learning this MNL mixture MDP equals to learning $H/2$ logistic bandit problems. This finishes the proof. ∎

# H Supporting Lemmas

In this section, we provide several supporting lemmas used in the proofs of the main results.

**Lemma 12** (Abbasi-Yadkori et al. [2011, Theorem 1]). *Let $\{\mathcal{F}_t\}_{t=0}^{\infty}$ be a filtration. Let $\{\eta_t\}_{t=1}^{\infty}$ be a real-valued stochastic process such that $\eta_t$ is $\mathcal{F}_t$-measurable and $\eta_t$ is conditionally zero-mean $R$-sub-Gaussian for some $R \geq 0$ i.e. $\forall \lambda \in \mathbb{R}, \mathbb{E}\left[e^{\lambda \eta_t} \mid F_{t-1}\right] \leq \exp\left(\lambda^2 R^2 / 2\right)$. Let $\{X_t\}_{t=1}^{\infty}$ be an $\mathbb{R}^d$-valued stochastic process such that $X_t$ is $\mathcal{F}_{t-1}$-measurable. Assume that $V$ is a $d \times d$ positive definite matrix. For any $t \geq 1$, define*

$$V_t = V + \sum_{s=1}^{t-1} X_s X_s^{\top}, \quad S_t = \sum_{s=1}^{t-1} \eta_s X_s.$$

*Then, for any $\delta \in (0,1)$, with probability at least $1 - \delta$, for all $t \geq 1$,*

$$\|S_t\|_{V_t^{-1}} \leq R\sqrt{2\log\left(\frac{\det(V_t)^{1/2}\det(V)^{-1/2}}{\delta}\right)}.$$

**Lemma 13** (Périvier and Goyal [2022, Theorem 4]). *Let $\{\mathcal{F}_t\}_{t=0}^{\infty}$ be a filtration. Let $\{\delta_t\}_{t=1}^{\infty}$ be an $\mathbb{R}^N$-valued stochastic process such that $\delta_t$ is $\mathcal{F}_t$-measurable one-hot vector. Furthermore, assume $\mathbb{E}[\delta_t|\mathcal{F}_{t-1}] = p_t$ and define $\varepsilon_t = p_t - \delta_t$. Let $\{X_t\}_{t=1}^{\infty}$ be a sequence of $\mathbb{R}^{N \times d}$-valued stochastic process such that $X_t$ is $\mathcal{F}_{t-1}$-measurable and $\|X_{t,i}\|_2 \leq 1, \forall i \in [N]$. Let $\{\lambda_t\}_{t=1}^{\infty}$ be a sequence of non-negative scalars. Define*

$$H_t = \sum_{i=1}^{t-1}\left(\sum_{j=1}^{N} p_j X_{i,j} X_{i,j}^{\top} - \sum_{j=1}^{N}\sum_{k=1}^{N} p_j p_k X_{i,j} X_{i,k}^{\top}\right) + \lambda_t I_d, \quad S_t = \sum_{i=1}^{t-1}\sum_{j=1}^{N} \varepsilon_{i,j} X_{i,j}.$$

*Then, for any $\zeta \in (0,1)$, with probability at least $1 - \zeta$, for all $t \geq 1$,*

$$\|S_t\|_{H_t^{-1}} \leq \frac{\sqrt{\lambda_t}}{4} + \frac{4}{\sqrt{\lambda_t}}\log\left(\frac{2^d \det(H_t)^{\frac{1}{2}} \lambda_t^{-\frac{d}{2}}}{\zeta}\right).$$

**Lemma 14** (Abbasi-Yadkori et al. [2011, Lemma 10]). *Suppose $x_1, \ldots, x_t \in \mathbb{R}^d$ and for any $1 \leq s \leq t$, $\|x_s\|_2 \leq L$. Let $V_t = \lambda I_d + \sum_{s=1}^{t} x_s x_s^{\top}$ for $\lambda \geq 0$. Then, we have $\det(V_t) \leq \left(\lambda + tL^2/d\right)^d$.*

**Lemma 15** (Orabona [2019, Lemma 6.9]). *Let $Z$ be a positive define matrix and $\mathcal{W}$ be a convex set, define $\mathbf{w}_{t+1}$ as the solution of*

$$\mathbf{w}_{t+1} = \arg\min_{\mathbf{w} \in \mathcal{W}}\left\{\langle \mathbf{g}, \mathbf{w}\rangle + \frac{1}{2\eta}\|\mathbf{w} - \mathbf{w}_t\|_Z^2\right\}.$$

*Then we have*

$$\|\mathbf{w}_{t+1} - \mathbf{w}_t\|_Z \leq 2\eta\|\mathbf{g}\|_{Z^{-1}}.$$

**Lemma 16** (Campolongo and Orabona [2020, Proposition 4.1]). *Define $\mathbf{w}_{t+1}$ as the solution of*

$$\mathbf{w}_{t+1} = \arg\min_{\mathbf{w} \in \mathcal{V}}\left\{\eta \ell_t(\mathbf{w}) + \mathcal{D}_{\psi}(\mathbf{w}, \mathbf{w}_t)\right\},$$

*where $\mathcal{V} \subseteq \mathcal{W} \subseteq \mathbb{R}^d$ is a non-empty convex set. Further supposing $\psi(\mathbf{w})$ is 1-strongly convex w.r.t. a certain norm $\|\cdot\|$ in $\mathcal{W}$, then there exists a $\mathbf{g}_t' \in \partial \ell_t(\mathbf{w}_{t+1})$ such that*

$$\langle \eta_t \mathbf{g}_t', \mathbf{w}_{t+1} - \mathbf{u}\rangle \leq \langle \nabla\psi(\mathbf{w}_t) - \nabla\psi(\mathbf{w}_{t+1}), \mathbf{w}_{t+1} - \mathbf{u}\rangle$$

*for any $\mathbf{u} \in \mathcal{W}$.*

**Lemma 17** (Lee and Oh [2024, Lemma D.3]). *Define $Q : \mathbb{R}^K \to \mathbb{R}$, such that for any $\mathbf{u} = (u_1, \ldots, u_K) \in \mathbb{R}^K, Q(\mathbf{u}) = \sum_{i=1}^{K} \frac{\exp(u_i)}{v + \sum_{k=1}^{K} \exp(u_k)}$. Let $p_i(\mathbf{u}) = \frac{\exp(u_i)}{v + \sum_{k=1}^{K} \exp(u_k)}$. Then, for all $i \in [K]$, we have*

$$\left|\frac{\partial^2 Q}{\partial i \partial j}\right| \leqslant \begin{cases} 3p_i(\mathbf{u}) & \text{if } i = j, \\ 2p_i(\mathbf{u})p_j(\mathbf{u}) & \text{if } i \neq j. \end{cases}$$

