# OpenReview forum: "Provably Efficient Reinforcement Learning with Multinomial Logit Function Approximation"
_NeurIPS.cc/2024/Conference — NeurIPS 2024 poster_

### Official Review · Reviewer_tHPh · 2024-06-16

**Soundness:** 3
**Presentation:** 3
**Contribution:** 2
**Rating:** 5
**Confidence:** 3

**Summary:**

This paper considers MDPs employing the MNL function for transition probability, following Hwang and Oh [2023]. The authors suggest efficient algorithms based on online Newton steps, inspired by [Hazan et al., 2014; Zhang et al., 2016; Oh and Iyengar, 2021]. Furthermore, to improve $\kappa$ dependency, they provide algorithms employing local learning with mirror descent inspired by [Zhang and Sugiyama, 2023; Lee and Oh, 2024]. The algorithms achieve $1/\sqrt{\kappa}$ or even detach the dependency of $\kappa$ from the leading term.

**Strengths:**

The suggested algorithms are computationally efficient and show improvement in $\kappa$ compared to the previous work of Hwang and Oh [2023].

**Weaknesses:**

- Their suggested algorithms do not seem novel because they are based on previously proposed methods for logistic or MNL bandits. Specifically, the online Newton update is widely studied for MNL or logistic bandits [Oh and Iyengar, 2021; Zhang and Sugiyama, 2023].

- Furthermore, the improvement on $\kappa$ is based on the mirror descent algorithm proposed in [Zhang and Sugiyama, 2023; Lee and Oh, 2024], and the proofs seem to follow the steps in [Zhang and Sugiyama, 2023; Lee and Oh, 2024] in the appendix.

- Lastly, the MNL model for transition probability may have an inherent weakness: the number of achievable states for each (k,n) must be finite, and it is required to know the state space of $S_{k,n}$.

[1] Faury, Louis, et al. "Jointly efficient and optimal algorithms for logistic bandits." International Conference on Artificial Intelligence and Statistics. PMLR, 2022.

**Questions:**

Are there any non-trivial technical novelties in utilizing the online mirror descent method for MDPs?

**Limitations:**

The authors discuss some interesting future work regarding regret bound. Additionally, I believe, as mentioned in Weaknesses, the MNL model for transition probability has an inherent weakness: the number of achievable states for each (k,n) must be finite, and it is required to know the state space of $S_{k,n}$.

---

> ### Author Rebuttal · Authors · 2024-08-06
>
> Thank you for your comment. We will address your questions and clarify any misunderstandings, which may be due to our inadequate emphasis on the technical contributions in the presentation. We will improve the clarity in the revised version.
>
> ---
>
> **Q1:** "Their suggested algorithms do not seem novel because they are based on previously proposed methods for logistic or MNL bandits... "
>
> **A1:** We respectively disagree with this claim. While the idea of online updates has been utilized in previous works, we make several novel contributions in the MDP setting:
>
> - Oh & Iyengar (2021) use the global information of the Hessian matrix, which fails to remove the dependence on $\kappa$. Moreover, we identify and address a technical issue present in their work as discussed in Remark 1, which is valuable to the community.
> - Zhang & Sugiyama (2023) consider a *multi-parameter* model (i.e., the unknown parameter is a matrix $W_h^\star$), which differs fundamentally from our *single-parameter* model (i.e., the unknown parameter is a vector $\theta_h^\star$). The techniques in Zhang & Sugiyama (2023) cannot be directly applied to our setting because some key properties of the functions differ. For instance, a direct application of their method would result in a regret bound that scales linearly with the number of reachable states.
> - Lastly, we establish the first lower bound for this problem, which is a novel technical contribution.
>
> To summarize, our paper presents *novel* algorithms and regret guarantees for RL with multinominal logit function approximation, and also contributes *novel* technical ingredients to achieve desired results.
>
> ---
>
> **Q2:** "The improvement on $\kappa$ is based on the mirror descent algorithm proposed in [Zhang & Sugiyama, 2023; Lee & Oh, 2024]. Are there any non-trivial technical novelties?"
>
> **A2:** While our work builds upon these two important prior works, our solution incorporates non-trivial technical innovations crucial for achieving favorable regret bounds, especially due to the concerns of the intrinsic dimension in RL with function approximation setting. Below, we highlight the technical novelties:
>
> - **Compared with Zhang & Sugiyama (2023).** As discussed in A1, Zhang & Sugiyama (2023) consider a *multi-parameter* model, which differs fundamentally from our *single-parameter* model. As a result, the techniques in Zhang & Sugiyama (2023) cannot be directly applied to our setting because some key properties of the functions differ.  A naive application of their method would lead to a regret bound that scales linearly with the number of the reachable states $U$.
> - **Compared with Lee & Oh (2024).** Lee & Oh (2024) study the single-parameter MNL bandit setting, and while our parameter update approach shares similarities with theirs, our construction of the optimistic value function differs significantly. Specifically, they use a first-order upper bound for each item, which is insufficient to remove the dependence on $\kappa$ in our setting. In contrast, we employ the second-order Taylor expansion to achieve a more accurate approximation of the value function. This approach is non-trivial and requires a different analysis.
>
> We will emphasize these technical novelties compared to prior works more clearly in the revised version.
>
> ---
>
> **Q3:** "The MNL model for transition probability may have an inherent weakness... "
>
> **A3:** It is a mild condition that the reachable state space is finite and known to the learner, which has been used in prior works. Below we illustrate the reasons:
>
> - This condition holds for many practical applications, such as the SuperMario game, where, despite the vast state space, the reachable state space is limited to four states and known to the learner, corresponding to the agent's possible movements: up, down, left, or right.
> - In fact, even for linear mixture MDPs, which are extensively studied in the literature (Zhou et al., 2021; He et al., 2022), the standard assumption is that $\sum_{s'} \psi(s') V(s')$ can be evaluated by an Oracle. This assumption assumes that the state space is finite and known to the learner implicitly. Moreover, several works on linear mixture MDPs have regret bounds that depend on the size of the state space (e.g., Zhao et al., 2023; Ji et al., 2024). This also implies that the state space is finite.
> - Moreover, even without exact information of $S_{k,h}$, our results also hold when replacing it with its upper bound.
>
> Importantly, the MNL model ensures valid distributions over states, addressing the limitations of linear function approximation. We believe this is a key shining feature of the MNL model, which will take an important step towards bridging the gap between theory and practice.
>
> ----
>
> We hope the above responses sufficiently address your concerns. If our responses have properly resolved the issues raised, we would appreciate it if you could consider re-evaluating our work. We're also happy to provide further clarifications to additional questions.
>
> ---
>
> **References:**
>
> [1] Zhou, D., Gu, Q., & Szepesvari, C., Nearly Minimax Optimal Reinforcement Learning for Linear Mixture Markov Decision Processes. In COLT'21.
>
> [2] He J., Zhou D., & Gu Q., Near-optimal Policy Optimization Algorithms for Learning Adversarial Linear Mixture MDPs. In AISTATS'22.
>
> [3] Zhao, C., Yang, R., Wang, B., & Li, S., Learning Adversarial Linear Mixture Markov Decision Processes with Bandit Feedback and Unknown Transition. In ICLR'23.
>
> [4] Ji, K., Zhao, Q., He, J., Zhang, W., & Gu, Q., Horizon-free Reinforcement Learning in Adversarial Linear Mixture MDPs. In ICLR'24.

---

> > ### Comment · Reviewer_tHPh · 2024-08-08
> > **Thank you for your response.**
> >
> > For the technical novelty, you mentioned that "In contrast, we employ the second-order Taylor expansion to achieve a more accurate approximation of the value function. This approach is non-trivial and requires a different analysis." Could you provide more details regarding this?

---

> > > ### Author Response · Authors · 2024-08-09
> > > **Thank you for your response.**
> > >
> > > Thank you for your response. We would like to take this opportunity to provide additional clarification on the technical contributions of our work.
> > >
> > > While our algorithm and analysis share certain similarities with existing work on bandits (e.g., Zhang and Sugiyama, 2023; Lee and Oh, 2024), our approach introduces several novel elements specifically tailored to the MDP setting.
> > >
> > > As the reviewer rightly pointed out (Point 2 of Weaknesses), the online estimation step and the analysis in Section 5.1 do draw from recent advances in bandits. However, it’s important to note that this step only mitigates the dependence to $O(\kappa^{-1/2})$, and an $O(\kappa^{-1/2})$ dependence still persists in the regret bound as shown in Theorem 2.
> > >
> > > To further reduce the $O(\kappa^{-1/2})$ dependence, it is crucial to examine how local information is preserved in the regret analysis. To this end, Zhang and Sugiyama (2023) use a second-order Taylor expansion to construct the optimistic revenue (Proposition 1), but their approach is designed for a multi-parameter model, which differs fundamentally from our single-parameter model. Lee and Oh (2024) employ *a first-order upper bound* for each item and *specific properties of MNL bandit* to construct the optimistic expected revenue (Eq. (5) and (6) of their paper), which does not hold in our setting.
> > >
> > > To preserve local information in our work, we need to estimate the value function more accurately. Specifically, whereas Lemma 2 addresses value differences between the value functions, Lemma 4 refines this by incorporating a second-order Taylor expansion, **allowing us to maintain the local information $p(\theta)$ rather than applying the maximum operator as in Lemma 2**. Although this second-order Taylor expansion is inspired by the work of Zhang and Sugiyama (2023), we need new analysis to address the challenges unique to the single-parameter model, such as handling the negative term in our first-order term, which is not present in their work. Moreover, a naive application of their analysis would lead to a regret bound that scales linearly with the number of the reachable states $U$, which is undesired in the MDP setting.
> > >
> > > We hope this clarifies the contributions of our approach. We are happy to provide further details if needed.

---

> ### Comment · Reviewer_tHPh · 2024-08-10
> **Thank you for your comment.**
>
> Overall, I agree that there are several adjustments needed to apply the mirror descent approach to RL. This work extends previous research on bandits to the RL setting. However, I have concerns about the significance of the technical novelty of this work. I believe that the second-order Taylor expansion mentioned by the author may not be unique to RL, as it is also used in bandit problems. Therefore, I'm maintaining my score.

---

> > ### Author Response · Authors · 2024-08-12
> > **Thanks for your reply.**
> >
> > We appreciate your feedback but have to disagree with this comment. We would like to offer some clarifications that underscore the contributions of our work to the field of RL.
> >
> > **Technical relationship between bandits and MDPs.** Since bandits can be viewed as one-step MDPs, it's *common and reasonable* for methodologies and analyses in MDPs to draw inspiration from bandits (e.g., linear bandits vs. linear MDPs). To some extent, bandit techniques are fundamental to modern RL theory and many RL algorithms are built upon bandit algorithms. However, the most intriguing and challenging aspect of RL theory lies in leveraging its unique structure, particularly when addressing the intrinsic dimension in function approximation settings.
> >
> > **Our unique technical challenge and innovations.** Our work builds on recent advancements in MNL bandits, which have been well acknowledged in our paper. However, it is crucial to note that *several non-trivial technical innovations are necessary* to achieve favorable regret bounds. While employing high-level ideas like the Taylor expansion is not entirely new, the corresponding terms of MNL MDPs are significantly different from the bandit setting, which requires *new and more sophisticated analyses*.
> >
> > **Significance and Impact of Our Results.** Beyond technical novelty, our results hold significant value and are important for the community. Understanding function approximation is one of the central challenges in RL theory and there are many efforts devoted to, yielding fruitful results. However, one limitation is the linear assumption may not guarantee valid transition probabilities. MNL MDPs address this limitation by incorporating non-linear function approximation, which brings significant challenges as well. Our work not only is *the first to achieve nearly the same statistical and computational efficiency* as linear function approximation but also establishes *the first lower bound* for this problem. Our work broadens the scope of function approximation greatly and makes a significant step forward in RL theory.
> >
> > We hope this clarification highlights the uniqueness and importance of our contributions and addresses your concerns regarding the novelty and impact of our work. We believe our work is crucial for the community. We appreciate it if you could reconsider your evaluation in light of these clarifications.

---

> > > ### Comment · Reviewer_tHPh · 2024-08-12
> > > **Thank you for your comment**
> > >
> > > Thank you for clarifying your contribution. I still believe that the technical contribution may not be significant, which is an important factor for this conference. This is why I consider the contribution of this work to be on the borderline (4 or 5). However, I may have overlooked the first contribution of this paper regarding the extension to RL theory using recent bandit algorithms and techniques. From that perspective, I have raised my score to 5.

---

> > > > ### Author Response · Authors · 2024-08-13
> > > > **Thank you.**
> > > >
> > > > Thanks for your re-evaluation! We will carefully revise the paper to better highlight the technical contributions and the significance of our results.

---

### Official Review · Reviewer_Roix · 2024-07-11

**Soundness:** 4
**Presentation:** 3
**Contribution:** 3
**Rating:** 6
**Confidence:** 3

**Summary:**

In this paper, the author analyzes a Markov Decision Process (MDP) model with non-linear function approximation. Specifically, in the finite-time horizon inhomogeneous episodic MDPs setting, the transition dynamics are unknown but the reward function is known. The author proposes using a multinomial logit (MNL) function approximation to estimate transition dynamics, which is superior to the linear function approximation if the model is misspecified in \cite{hwang2023model}. Additionally, the author proposes *UCRL-MBL-OL*, which adapts the previous work that is model-based and has large computational and storage complexity, to an online style that only consumes constant computation and storage resources. Moreover, the author has proven that the regret bound of *UCRL-MBL-OL* matches the state-of-the-art in Theorem 1. Its regret bound achieves $\tilde{O}(\kappa^{-1} dH^2\sqrt{K})$, where $H$ is the time horizon length, $K$ is the number of total episodes and $\kappa$ is considered as a parameter to control the sparsity of the transition dynamics and $d$ is the hidden dimensionality. Ignoring the logarithmic factor and $\kappa$, such a regret bound has only a $\sqrt{H}$ gap compared to the lower bound. After that, with additional assumption, the author utilizes the local information to propose another two algorithms, *UCRL-MNL-LL* and *UCRL-MNL-LL+* to remove the dependence on $\kappa$ and get a tighter regret bound as well as maintain good properties of *UCRL-MNL-OL*.

**Strengths:**

1. This paper is well-written. The author makes a clear improvement point compared to the literature.

2. The algorithm proposed by the author enjoys an online learning style that does not need to maintain a large historical set.

**Weaknesses:**

1. Although this paper focuses on reducing the computation complexity, I am curious about the sample complexity of *UCRL-MNL-OL*.

2. Since the algorithm builds up the estimation of the transition dynamics by using MNL function approximation, is it considered a model-based algorithm? More specifically, does it require storing the transition dynamics for each state-action pair in every step?

**Questions:**

Please see the above "Weaknesses"

---

> ### Author Rebuttal · Authors · 2024-08-06
>
> Thank you for your helpful feedback. We will address your questions below.
>
> ---
>
> **Q1:** "Although this paper focuses on reducing the computation complexity, I am curious about the sample complexity of UCRL-MNL-OL."
>
> **A1:** Thanks for your question. This work not only reduces computational complexity but also significantly improves sample complexity. A sample complexity guarantee can be derived from any low-regret algorithm using an online-to-batch conversion, as demonstrated by Jin et al. (2019). Specifically, suppose we have total regret $\sum_{k=1}^K[V_1^{\star}(s_1)-V_1^{\pi_k}(s_1)] \leq \sqrt{CK}$, then, randomly selecting $\pi = \pi_k$ we have that $[V_1^{\star}(s_1)-V_1^{\pi}(s_1)] \leq \sqrt{C/K}$ with constant probability by Markov's inequality. Thus, Theorem 1 implies UCRL-MNL-OL has a sample complexity of $O(\kappa^{-1} d H^2 / \epsilon^2)$. We will add this discussion to the revised version.
>
> ---
>
> **Q2:** "Is it considered a model-based algorithm? Does it require storing the transition dynamics for each state-action pair in every step?"
>
> **A2:** Our algorithm is model-based because it first estimates the transition parameters and then derives the value function based on these parameters. However, unlike traditional model-based methods, it does not require storing the transition dynamics for each state-action pair at every step. Instead, we only store the estimated transition parameter (d-dimension vector) and compute the transitions at the current state as needed. This efficiency is achieved by updating the parameters in an online manner, eliminating the need to store the entire history of transitions.
>
> ---
>
> We hope that our responses have addressed your concerns. We will be happy to provide further clarification if needed.
>
> ---
> **References:**
>
> [1] Jin, C., Allen-Zhu, Z., Bubeck, S. & Jordan, I. M., Is Q-learning provably efficient? In NeurIPS'18.

---

> > ### Comment · Reviewer_Roix · 2024-08-13
> >
> > Thanks for your answers. It would be great to add sample complexity to the paper. I will not change the rating and will continue to support this work.

---

> > > ### Author Response · Authors · 2024-08-13
> > > **Thank you.**
> > >
> > > Thank you for your helpful suggestion. We will include a discussion on sample complexity in the next version.

---

### Official Review · Reviewer_xBZM · 2024-07-11

**Soundness:** 3
**Presentation:** 3
**Contribution:** 3
**Rating:** 6
**Confidence:** 4

**Summary:**

This work studies the MNL function approximation inhomogeneous RL, achieves the $O(1)$ computation cost, and improves the regret guarantee with regard to $\kappa$. To improve the computation cost, this work employs the online Newton step instead of MLE estimation to estimate $\theta$. Then, they design a novel confidence set by making full use of local information to improve the dependence of $\kappa$.

**Strengths:**

1.	The use of local information instead of a uniform $\kappa$ is novel and useful to improve the dependence of $\kappa$.
2.	The UCRL-MNL-LL+ removes the $\kappa$ dependence on the lower-order term and almost matches the optimal regret results by using high-order Taylor expansion.

**Weaknesses:**

1.	[1] also use the online Newton step to improve the computation cost in the logit contextual bandits setting. It would be better to discuss the novelty of UCRL-MNL-OL.

[1] Oh, M. H., & Iyengar, G. (2021, May). Multinomial logit contextual bandits: Provable optimality and practicality. In Proceedings of the AAAI conference on artificial intelligence (Vol. 35, No. 10, pp. 9205-9213).

**Questions:**

Question 1: This work achieves great results in the stochastic reward setting. Can you discuss the challenge when extending to the adversarial reward setting?

**Limitations:**

The authors adequately addressed the limitations.

---

> ### Author Rebuttal · Authors · 2024-08-06
>
> Thanks for your insightful review. We will address your questions below.
>
> ---
>
> **Q1:** "Oh & Iyengar (2021) also use the ONS to improve the computation cost..."
>
> **A1:** As mentioned in Line 184, the parameter estimation of UCRL-MNL-OL algorithm is inspired by the work of Oh & Iyengar (2021). However, to construct the optimistic value function efficiently in MNL-MDPs, we had to develop a different approach tailored to the specific structure of the value function in MNL-MDPs. Moreover, as highlighted in Remark 1 of our paper, we identify and address a technical issue present in their work.
>
> ---
>
> **Q2:** "Can you discuss the challenge when extending to the adversarial reward setting?"
>
> **A2:** Thanks for your insightful question. Extending our method to the adversarial reward in the full-information setting is relatively straightforward. Instead of using a greedy policy (i.e., $a = \mathrm{argmax}~ Q_k(s, \cdot)$) as in our current approach, we can incorporate a policy optimization step, such as $\pi_{k+1}(\cdot | s) = \pi_k(\cdot | s) \exp(\eta Q_k(s, \cdot))$. This modification would result in an additional regret term of $O(H^2 \sqrt{K})$. However, extending our method to the adversarial reward setting for the bandit setting is more challenging, as we need to handle both unknown transitions and unknown rewards simultaneously. To the best of our knowledge, even the state-of-the-art algorithms for linear mixture MDPs in this setting of Li et al. (2024) exhibit a regret bound that scales linearly with the number of states.
>
> ---
>
> We hope that our responses have addressed your concerns. We will be happy to provide further clarification if needed.
>
> ---
>
> **Reference:**
>
> [1] Li, L. F., Zhao, P., & Zhou, Z. H., Improved Algorithm for Adversarial Linear Mixture MDPs with Bandit Feedback and Unknown Transition. In AISTATS'24.

---

> > ### Comment · Reviewer_xBZM · 2024-08-12
> >
> > Thanks for your careful response. The rebuttal addresses my concerns and it would be better to add the above content in the next version. I maintain my score to support this valuable work.

---

> > > ### Author Response · Authors · 2024-08-14
> > > **Thanks for your reply.**
> > >
> > > Thank you for your helpful suggestions. We will incorporate these discussions into the next version. Thanks again!

---

### Official Review · Reviewer_zUWe · 2024-07-15

**Soundness:** 3
**Presentation:** 3
**Contribution:** 2
**Rating:** 6
**Confidence:** 3

**Summary:**

The problem considered in this paper is online learning in MDPs where transition probabilities are modelled with a log-linear model (with "multinomial logit function approximation"). The finite horizon, time-inhomogenous setting is considered. The problem is motivated by allowing a nonlinear transformation in modeling the MDP and yet maintaining both computational and information theoretic tractability. Inspired by results in the analogous bandit problems and algorithms developed for them, a number of gradually more complex, but (statistically) better performing algorithms are considered. In particular, while naive approaches give a poor dependence on a problem parameter $\kappa$ that characterizes the "strength" of nonlinearity, by adopting previous ideas to the MDP setting, new algorithms are designed that eliminate this poor dependence. A lower bound is also established, which nearly matches the upper bound (but considers infinite action spaces, while the main paper considers finite action spaces).

**Strengths:**

This is a reasonable problem setting; and the approach is also reasonable. It is nice to have a lower bound, even if there is a mismatch between the settings. It is nice to see that ideas that were developed for the bandit setting generalize to the MDP setting.

**Weaknesses:**

1. The novelty is limited by that we have seen the same story, same ideas playing out nicely in the closely related bandit setting.
2. A new parameter, U, the number of next states that are reachable with positive probability in the worst case, appears in the analysis and will appear in the bounds.
3. It is an unpleasant surprise for the reader to discover this dependence only through carefully reading the paper, rather than being told upfront. It is not good that the opportunity to discuss whether this quantity needs to enter the regret bound, and that this quantity needs to be small for the algorithm to be tractable, is missed.
4. Line 83 and onward: The work of Uohamma is discussed but is mischaracterized. My reading of this work is that they do establish that their algorithm runs in polynomial time. It remains unclear why the exponential family model is incomparable with the one considered here; an explanation (with examples) is missing.
5. The paper could use some extra proofreading (e.g., the upper indices in the bottom of page 5, in the displayed equation are not correct); in line 149, in the definition of $U$, $|\cdot|$ is missing.

**Questions:**

1. Can you confirm that the regret and compute cost depend on U, the worst-case number of next states that one can transition to with positive probability? Do you think such dependencies are necessary? Are there any interesting examples where it is reasonable to expect that U is small, independently of the size of the state space?
2. What was the most challenging aspect of extending the bandit ideas to the MDP framework?

**Limitations:**

n.a.

---

> ### Author Rebuttal · Authors · 2024-08-06
>
> Thanks for your constructive review. Below, we will address your main questions, especially regarding the dependence on $U$ (see A1-a,b,c), technical challenges (see A2), the difference to prior work (see A3), and presentation issues (see A4).
>
> ---
>
> **Q1-a:** "The regret and compute cost depend on $U$?....  Do you think such dependencies are necessary? "
>
> **A1-a:** We appreciate your observation. Though $\kappa$ may have polynomial dependence on $U$ in the worst case, the leading term of our best result (Theorem 3) is independent of $\kappa$ and only has a logarithmic dependence on $U$. While the lower-order terms do depend on $\kappa$, they are independent of the number of episodes $K$ and can be treated as constants. Thus, this dependence is believed to be acceptable. The dependence on $U$ is necessary for our method and we explain the reasons below.
>
> In general, methods for MDPs can be classified into two categories: model-based and model-free. Model-based methods aim to maximize the likelihood of observed data while model-free methods focus on minimizing the mean-squared error in predicting value functions.
> - The dependence on $U$ for regret and computational cost is usually necessary for the model-based methods, as it controls the model's error, which typically involves a total of $U$ elements. Similar dependencies have been observed in the literature (e.g., Hwang & Oh, 2023).
> - Most papers on linear MDPs use the value-targeted approach, a model-free method that does not rely on such dependencies. This is feasible because the linearity of the value function in linear MDPs allows for learning the value function directly. In contrast, **the value function for MNL-MDPs is neither linear nor log-linear**, preventing using model-free methods.
>
> We will further elaborate on this point in the revised version.
>
> ---
>
> **Q1-b:** "Examples where $U$ is small...?"
>
> **A1-b:** This phenomenon is quite common in practice, as in many applications, the agent tends to transit to nearby states even if the entire state space is large. An illustrative example is the RiverSwim environment in Hwang & Oh (2023), where although the state space is extensive, the agent only transits to adjacent states. Another example is the SuperMario game, where, despite a vast state space, the value of $U$ is limited to 4, corresponding to the possible movements: up, down, left, or right.
>
> ---
>
> **Q1-c:** "It is an unpleasant surprise for the reader to discover this dependence..."
>
> **A1-c**: Thanks for your helpful suggestion. We will discuss this dependence in the next version.
>
> ---
>
> **Q2:** "The novelty is limited... What was the most challenging aspect...?"
>
> **A2**: While similar ideas have been explored in the bandit setting, there are several unique challenges specific to MDPs that need to be addressed, especially due to the concerns of the intrinsic dimension in RL with function approximation setting.
>
> - **Compared with Zhang & Sugiyama (2023).** The model of Zhang & Sugiyama (2023) fundamentally differs from ours. We employ a *single-parameter* (i.e., a vector $\theta_h^*$) model for different states at each stage $h$, whereas they use a *multi-parameter* model (i.e., a matrix $W_h^*$). As a result, the techniques in Zhang & Sugiyama (2023) cannot be directly applied to our setting because some key properties of the functions differ. For instance, a direct application of their method would result in a regret bound that scales linearly with the number of reachable states $U$.
> - **Compared with Lee and Oh (2024).** While Lee and Oh (2024) study the single-parameter MNL bandit setting, and our parameter update approach shares some similarities to theirs, their construction of the optimistic value function is significantly different. They use a first-order upper bound for each item, which is insufficient to remove the dependence on $\kappa$ in our setting. To address this, we employ the second-order Taylor expansion to achieve a more accurate approximation of the value function.
>
> We will emphasize the challenges and our contributions more clearly in the next version.
>
> ---
>
> **Q3:** "The work of Ouhamma et al. (2023) and why it is incomparable.."
>
> **A3:** Thanks for pointing it out. Ouhamma et al. (2023) studied the exponential family model, a similar setting to ours. However, our work exhibits differences from theirs for both problem setting and computational complexity. Below, we make the clarifications.
>
> - **Problem setting.** The MNL MDPs studied in this paper can not be covered by the exponential family of MDPs in Ouhamma et al. (2023). An example of an MNL MDP that does not belong to the exponential family is given by the function $\phi_i(s, a, s') = \exp((s' - s -i)^/a^2) / \sqrt{\pi a^2}$. It does not belong to the exponential family as $\phi_i(s, a, s')$ can not be decomposed in the form $\phi_i(s, a) \cdot \psi_i(s')$. More discussion on this point can be found in Zhou et al. (2021).
> - **Computational complexity.** Though the algorithm of Ouhamma et al. (2023) runs in pseudo-polynomial time, our method is more efficient regarding storage and computational complexity. They estimate the transition parameters using the maximum likelihood estimation (MLE), which requires storing all previously observed data. Consequently, the computational complexity at episode $k$ is $O(k)$. In contrast, we employ an online parameter update method, which requires only $O(1)$ storage and computational complexity per episode.
>
> We will provide a more detailed comparison in the next version.
>
> ---
>
> **Q4:** "The paper could use some extra proofreading"
>
> **A4:** We appreciate your feedback and will ensure that the revised version is thoroughly proofread.
>
> ---
>
> We hope that our responses have addressed your concerns. We will be happy to provide further clarification if needed.
>
> ---
>
> **Reference:**
>
> [1] Zhou, D., He, J., & Gu, Q., Provably Efficient Reinforcement Learning for Discounted MDPs with Feature Mapping. In ICML'21.

---

> ### Comment · Reviewer_zUWe · 2024-08-13
>
> Thanks for the rebuttal; it was useful.
> Overall, my feelings towards the paper have not changed: This is a fine peace of work that should get published. I still have some reservations concerning the model; in fact, one of the reservations I have I forgot to include into my original review is that the algorithm needs to know the support of the next state distribution. This combined with that the size of this support appears in the bound, and that in the related linear kernel MDP setting, the size of the support does not appear in the bounds makes me feel uneasy about the paper. Why is knowing the support problematic? Well, if the size of the support was not part of the bound, one could just "max out" the support (all states are possible next states). If not, in the lack of results showing that the algorithm is robust in the face of misspecified next state support, one is afraid that knowing the support will actually be important. And I expect knowing the support is kinda tricky. Consider for example the SuperMario game, mentioned in the rebuttal. Here, knowing the next states (all 4 of them) encodes a tremendous amount of information about the game. Just consider how many states there are here and compare this to the number 4. How realistic is that one would actually know the support in scenarios like this. So I have doubts. I think an example, where the assumptions are more natural would tremendously help the paper. Also, why not think of some "real" application that people may care about (I have to say I have a hard time imagining anyone to seriously care about how to play SuperMario; and I also have a hard time imagining a more general problem that has the characteristics of SuperMario (ie I don't have a problem with simplified examples, but SuperMario and other games just feel like too arbitrary and unrelated to any "real world" applications).)
> In summary, notwithstanding these reservations, I am in support of accepting the paper, given that it looks at a somewhat reasonable setting and makes a nontrivial contribution.

---

> > ### Author Response · Authors · 2024-08-14
> > **Thanks for your valuable comment.**
> >
> > Thanks for your constructive feedback. We agree with the reviewer's observation that the support of the next state distribution plays a key role in the MNL MDP model. Investigating how to remove this prior knowledge is an important future direction.
> >
> > Even though, there are some real-world applications beyond games where the support of the next state distribution is limited and known. For instance, in the robot navigating problem, the robot can only move to nearby locations, even though the overall state space may be extensive. Similarly, in language models, the current state can be conceptualized as the sequence of previously generated words, with the immediately accessible next states being the potential subsequent words. Although the vocabulary could be extensive, the feasible choices for the next word, dictated by grammar and context, are inherently limited and known.
> >
> > We will add these discussions in the next version. We appreciate your helpful suggestions!

---

### Official Review · Reviewer_qUhc · 2024-08-05

**Soundness:** 3
**Presentation:** 3
**Contribution:** 3
**Rating:** 6
**Confidence:** 3

**Summary:**

The paper studies the recently proposed MDPs that use multinomial logit function approximation for state distribution validness. The results and algorithms improve the prior work of Hwang and Oh [2023] in multiple aspects, including computation efficiency, storage, and statistical dependence on the problem-dependent quantity $\kappa$ that can be exponentially small. In addition, the authors establish a matching lower bound on $d$, the feature space dimension, and $K$, the number of episodes.

**Strengths:**

- The paper is well-written and has clear logic flows. Readers can see how the authors approach the MDP problem and tackle the challenges. In particular, Table 1 is quite useful for demonstrating the advancements in the work.
- The improvements in both computation and storage efficiencies are essential for practical applications. In Theorem 2, the authors also improve the dependence on $kappa$ to $\sqrt{\kappa}$ without affecting efficiency. The enhancement seems significant, especially since the parameter can be exponentially small.
- The lower bound established in the paper is the first to demonstrate the optimality of the authors' algorithms in the $d$-$K$ dependence. Per my understanding, it also confirms the results' optimality of Hwang and Oh [2023].

**Weaknesses:**

- The primary high-level techniques and tools (seem to) come from existing works and relevant fields, such as MNL contextual bandits. The authors should put more effort into highlighting the technical challenges and novelties besides the previous comparisons.
- It would be beneficial to include experiments on synthetic and real-world datasets and compare the results to existing baselines and relevant works. In particular, the new algorithms seem more involved than prior ones, which may affect their stability and adaptiveness.
- There is still a significant gap between the lower and upper bounds. Besides, I wonder how often $\kappa$ could be exponentially small in practical settings, though it's definitely of theoretical interest to approach the lower limits on parameter dependency.

**Questions:**

Overall, I think the paper makes reasonable contributions to the problem, and I have no additional questions/comments besides the above.

**Limitations:**

The authors have made various comparisons and discussed the limitations of the results, which I'm satisfied with. I do not see any potential negative societal impact of their work.

---

> ### Author Rebuttal · Authors · 2024-08-06
>
> Thanks for your constructive review. We will address your concerns below.
>
> ---
>
> **Q1:** "The primary high-level techniques and tools (seem to) come from existing works and relevant fields..."
>
> **A1:** While similar ideas have been explored in the bandit setting, there are several unique challenges specific to MDPs that need to be addressed, especially due to the concerns of the intrinsic dimension in RL with function approximation setting. Below, we highlight the key differences between our work and the existing literature:
>
> - **Compared with Zhang & Sugiyama (2023).** The model of Zhang & Sugiyama (2023) fundamentally differs from ours. We employ a *single-parameter* (i.e., a vector $\theta_h^*$) model for different states at each stage $h$, whereas they use a *multi-parameter* model (i.e., a matrix $W_h^*$). As a result, the techniques in Zhang & Sugiyama (2023) cannot be directly applied to our setting because some key properties of the functions differ. For instance, a direct application of their method would result in a regret bound that scales linearly with the number of reachable states $U$.
> - **Compared with Lee and Oh (2024).** While Lee and Oh (2024) study the single-parameter MNL bandit setting, and our parameter update approach shares some similarities to theirs, their construction of the optimistic value function is significantly different. They use a first-order upper bound for each item, which is insufficient to remove the dependence on $\kappa$ in our setting. To address this, we employ the second-order Taylor expansion to achieve a more accurate approximation of the value function.
>
> We will emphasize the challenges and our contributions more clearly in the revised version.
>
> ---
>
> **Q2:** "It would be beneficial to include experiments on synthetic and real-world datasets..."
>
> **A2:** Thanks for your suggestions! We agree that adding experiments would be beneficial to the work. Nevertheless, our primary goal is to enhance the theoretical understanding of RL function approximation rather than designing a specific algorithm with state-of-the-art empirical performance for MNL MDPs. More specifically, the main focus of our paper is to investigate *whether we can achieve (almost) the same computational and statistical efficiency as linear function approximation while employing a more expressive non-linear function approximation*. We answer this question affirmatively by considering the MNL MDPs, which broaden the scope of RL with function approximation. Therefore, to some extent, our focus is mainly on the theoretical aspect. Given that our current upper bounds still exhibit a gap compared to the lower bound (as discussed in the next question), we may have to focus on how to achieve the minimax optimal rate for now, which is definitely a challenging open problem to work with and leave the empirical evaluation as the future work.
>
> ---
>
> **Q3:** "There is still a significant gap between the lower and upper bounds."
>
> **A3:** There is indeed a gap between the lower and upper bounds, but may not be as significant as it seems. The lower bound we established is *instance-dependent*, and if we focus on the worst-case guarantee, the lower bound is $\Omega(d H \sqrt{K})$. Our upper bound only looses a factor of $H$ compared to the lower bound, which is acceptable. As we discussed in Line 306-314, closing this gap remains a challenging open problem, and we will add more discussion on this point in the revised version.
>
> ---
>
> **Q4:** "I wonder how often $\kappa$ could be exponentially small..."
>
> **A4:** Thanks for your insightful question. The phenomenon that $\kappa$ is exponentially small is quite common in practice. As $\kappa$ is defined as minimum value of the product of any two state transition probabilities (i.e., $\inf_{\theta \in \Theta} p_{s, a}^{s^{\prime}}(\theta) p_{s, a}^{s^{\prime \prime}}(\theta) \geq \kappa$), it will be extremely small if there are some hard-to-reach states with very low transition probabilities. For example, in autonomous driving, there are emergency states that are rare and have very low transition probabilities. Similarly, in financial trading, sudden market crashes or booms are rare events that can be considered hard-to-reach states in the market conditions state space. These events typically occur under unusual conditions and are not frequently observed, resulting in low transition probabilities.
>
> ---
>
> We hope our responses address your concerns. We are happy to provide further clarification if needed. Thanks again for your valuable feedback.

---

### Decision · Program_Chairs · 2024-09-25

**Decision:**

Accept (poster)

**Comment:**

This work makes multiple contributions on bounding the regret in the recently introduced problem of reinforcement learning where the transition kernel is modeled as a multinomial logit. On the positive side, a first result obtains the same regret as an earlier result of Hwang and Oh but with much better efficiency in terms of space and computation. A next result greatly reduces the order of dependence on a commonly appearing, often very small problem-dependent constant ($\kappa$). A third result entirely avoids dependence on $\kappa$ in the leading term although paying linearly in $\kappa^{-1}$ in a non-leading term. Also, a lower bound from a somewhat related setting (but not identical, as the lower bound considers a setting with infinite action space) suggests the authors’ results are nearly optimal.

From a technical standpoint, this work seems to be reasonably strong (maybe not very strong, but in light of the contributions, it seems nice).

Another positive point which I only got from closely looking at the paper is that the authors identify technical issues in previous analyses and explain how to fix them. This is nice.

With the above said, there are some strong reservations about this work which should be very well addressed in the next revision:
- First, the regret bounds depend on $U$, the maximum number of reachable states. This dependence was not made clear in the presentations in the theorems in the main text, and the paper lacked discussion about this new dependence. In the discussion period, the authors argued that in many situations, $\kappa$ may already have some dependence on $U$. However, being up front about dependencies on problem-dependent constants, and discussing them, is very important.
- The authors’ work requires knowing the support of the next-state distribution. From the author discussion period, the examples provided by the authors to justify having knowledge of the support were not that convincing.

Overall, this is a worthwhile paper that would fit well in the conference proceedings. Despite the reservations stated above, I believe the paper should be accepted.